# Role of N343 glycosylation on the SARS-CoV-2 S RBD structure and co-receptor binding across variants of concern

Callum M Ives[1†], Linh Nguyen[2†], Carl A Fogarty[1], Aoife M Harbison[1], Yves Durocher[3,4], John Klassen[2*], Elisa Fadda[5*]

[1]Department of Chemistry, Maynooth University, Maynooth, Ireland; [2]Department of Chemistry, University of Alberta, Edmonton, Canada; [3]Human Health Therapeutics Research Centre, Life Sciences Division, National Research Council Canada, Québec, Canada; [4]Département de Biochimie et Médecine Moléculaire, Université de Montréal, Québec, Canada; [5]School of Biological Sciences, University of Southampton, Southampton, United Kingdom

*For correspondence:
john.klassen@ualberta.ca (JK);
elisa.fadda@soton.ac.uk (EF)

[†]These authors contributed equally to this work

Competing interest: The authors declare that no competing interests exist.

**Abstract** Glycosylation of the SARS-CoV-2 spike (S) protein represents a key target for viral evolution because it affects both viral evasion and fitness. Successful variations in the glycan shield are difficult to achieve though, as protein glycosylation is also critical to folding and structural stability. Within this framework, the identification of glycosylation sites that are structurally dispensable can provide insight into the evolutionary mechanisms of the shield and inform immune surveillance. In this work, we show through over 45 μs of cumulative sampling from conventional and enhanced molecular dynamics (MD) simulations, how the structure of the immunodominant S receptor binding domain (RBD) is regulated by N-glycosylation at N343 and how this glycan's structural role changes from WHu-1, alpha (B.1.1.7), and beta (B.1.351), to the delta (B.1.617.2), and omicron (BA.1 and BA.2.86) variants. More specifically, we find that the amphipathic nature of the N-glycan is instrumental to preserve the structural integrity of the RBD hydrophobic core and that loss of glycosylation at N343 triggers a specific and consistent conformational change. We show how this change allosterically regulates the conformation of the receptor binding motif (RBM) in the WHu-1, alpha, and beta RBDs, but not in the delta and omicron variants, due to mutations that reinforce the RBD architecture. In support of these findings, we show that the binding of the RBD to monosialylated ganglioside co-receptors is highly dependent on N343 glycosylation in the WHu-1, but not in the delta RBD, and that affinity changes significantly across VoCs. Ultimately, the molecular and functional insight we provide in this work reinforces our understanding of the role of glycosylation in protein structure and function and it also allows us to identify the structural constraints within which the glycosylation site at N343 can become a hotspot for mutations in the SARS-CoV-2 S glycan shield.

## eLife assessment

This study presents an **important** finding on the structural role of glycosylation at position N343 of the SARS-CoV-2 spike protein's receptor-binding domain in maintaining its stability, with implications across different variants of concern. The evidence supporting the claims of the authors is **convincing**, since appropriate and validated methodology in line with current state-of-the-art has been approached. The work will be of interest to evolutionary virologists.

**Figure 1.** Structure of the SARS-CoV-2 (WHu-1) Spike (S) glycoprotein and of the receptor binding domain (RBD) with a heat map of the interactions between the N343 glycan and the RBD in different variants of concern. (**a**) Atomistic model of the SARS-CoV-2 (WHu-1) S glycoprotein trimer embedded in a lipid bilayer as reported in *Casalino et al., 2020*. In the conformation shown, the S bears the receptor binding domain (RBD) of chain A in an open conformation, highlighted with a solvent accessible surface rendering. The topological S1 and S2 subdomains are indicated on the left-hand side. Glycans are represented with sticks in white, the protein is represented with cartoon rendering with different shades of cyan to highlight the chains. (**b**) Close-up of the open RBD (WHu-1) in a angiotensin-converting enzyme 2 (ACE2)-bound conformation (PDB 6M0J), with regions colour-coded as described in the legend. Key residues for anchoring the N343 glycan (GlyTouCan-ID G00998NI; FA2G2 Oxford nomenclature), namely S371, S373, and S375, across the beta sheet core are highlighted also in the symbol nomenclature for glycans (SNFG) diagram on the bottom-right with links to the monosaccharides corresponding to primary contacts. Key residues of the hydrophobic patch (orange) found to be inverted in the recently isolated FLip XBB1.5 variant are also indicated. (**c**) Heat map indicating the interactions frequency (%) classified in terms of hydrogen bonding and van der Waals contacts between the N343 glycan and the RBD residues 365–375 for each variants of concern (VoC), over the cumulative conventional MD (cMD) and enhanced gaussian accelerated MD (GaMD) sampling. (**d**) Side view of the RBD highlighting the GM1o binding region (SNFG colouring) and the antigenic Region 1 (green s), Region 2 (or RBM in ice-blue), and Region 3 (orange). Key residues Y351, L452, T470, and E484 are labelled and shown with sticks. N-glycans (white sticks) are labelled according to their linkage to residues Asn 331 and Asn 343. Rendering done with VMD (https://www.ks.uiuc.edu/Research/vmd/).

## Introduction

The SARS-CoV-2 spike (S) glycoprotein is responsible for viral fusion with the host cell, initiating an infection that leads to COVID-19 (*Walls et al., 2020*; *Wrapp et al., 2020*). S is a homotrimer with a structure subdivided into two topological domains, namely S1 and S2, see *Figure 1a*, separated by a furin site, which is cleaved in the pre-fusion architecture (*Walls et al., 2020*; *Wrapp et al., 2020*). In the Wuhan-Hu-1 strain (WHu-1), and still in most variants of concern (VoCs), host cell fusion is predominantly triggered by S binding to the angiotensin-converting Enzyme 2 (ACE2) receptor located on the host cell surface (*Jackson et al., 2022*; *Wrapp et al., 2020*). This process is supported by glycan co-receptors, such as heparan sulfate (HS) in the extracellular matrix (*Clausen et al., 2020*; *Kearns et al., 2022*) and by monosialylated gangliosides oligosaccharides (GM1os and GM2os) peeking from the surface of the host cells (*Nguyen et al., 2022*). The interaction with ACE2 requires a dramatic conformational change of the S, known as 'opening,' where one or more receptor binding domains (RBDs) in the S1 subdomain become exposed. The region of the RBD in direct contact with the ACE2 surface is known as receptor binding motif (RBM) (*Jackson et al., 2022*; *Lan et al., 2020*; *Yi et al., 2020*). Ultimately, S binding to ACE2 causes shedding of the S1 subdomain and the transition to a

post-fusion conformation, which exposes the fusion peptide near the host cell surface, leading to viral entry (*Dodero-Rojas et al., 2021*; *Jackson et al., 2022*).

To exert its functions, S sticks out from the viral envelope where it is exposed to recognition. To evade the host immune system, enveloped viruses hijack the host cell's glycosylation machinery to cover S with a dense coat of host carbohydrates, known as a glycan shield (*Casalino et al., 2020*; *Chawla et al., 2022*; *Grant et al., 2020*; *Turoňová et al., 2020*; *Watanabe et al., 2020b*; *Watanabe et al., 2019*). In SARS-CoV-2 the glycan shield screens effectively over 60% of the S protein surface (*Casalino et al., 2020*), leaving the RBD, when open, and regions of the N-terminal domain (NTD) vulnerable to immune recognition (*Bangaru et al., 2022*; *Carabelli et al., 2023*; *Chawla et al., 2022*; *Chen et al., 2023*; *Harvey et al., 2021*; *Piccoli et al., 2020*). The RBD targeted by approximately 90% of serum neutralising antibodies (*Piccoli et al., 2020*) and thus a highly effective model not only to screen antibody specificity (*Du et al., 2020*; *Lan et al., 2020*; *Lin et al., 2022*) and interactions with host cell co-receptors (*Clausen et al., 2020*; *Mycroft-West et al., 2020*; *Nguyen et al., 2022*), but also as a protein scaffold for COVID-19 vaccines (*Dickey et al., 2022*; *Kleanthous et al., 2021*; *Montgomerie et al., 2023*; *Ochoa-Azze et al., 2022*; *Tai et al., 2020*; *Valdes-Balbin et al., 2021*; *Yang et al., 2022*).

As a direct consequence, the RBD is under great evolutionary pressure. Mutations of the RBD leading to immune escape are particularly concerning (*Cao et al., 2023*; *Starr et al., 2021*), especially when such changes enhance the binding affinity for ACE2 or give access to alternative entry routes (*Baggen et al., 2023*; *Cervantes et al., 2023*). The identification of mutational hotspots (*Cao et al., 2023*) and the effects of mutations in and around the RBM have been and are under a great deal of scrutiny (*Starr et al., 2022a*; *Starr et al., 2022b*; *Bloom et al., 2023*; ; *Barton et al., 2021*). Yet, less attention is devoted to mutations in the glycan shield, which have been shown to lead to dramatic changes in infectivity (*Harbison et al., 2022*; *Kang et al., 2021*; *Zhang et al., 2022*) and in immune escape (*Newby et al., 2023*; *Pegg et al., 2023*). Successful changes in the glycan shield are evolutionarily difficult to achieve, since the nature and pattern of glycosylation of the S is crucial not only to the efficiency of viral entry and evasion, but also to facilitate folding and to preserve the structural integrity of the functional fold. Therefore, identifying potential evolutionary hotspots in the S shield is a complex matter, yet of crucial importance to immune surveillance.

Potential changes of the shield, e.g., loss, shift, or gain of new glycosylation sites, can likely occur only where these do not negatively impact the integrity of the underlying, functional protein architecture. In this work, we present and discuss the case of N343, a key glycosylation site on the RBD. Results of extensive sampling from molecular dynamics (MD) simulations exceeding 45 μs, show how the loss of *N*-glycosylation at N343 affects the structure, dynamics, and co-receptor binding of the RBD, and how these effects are modulated by mutations in the underlying protein, going from the WHu-1 strain through the VoCs designated as alpha (B.1.1.7), beta (B.1.351), delta (B.1.617.2), and omicron (BA.1). In addition, we provide important insight into the structure and dynamics of the omicron BA.2.86 RBD. This variant, designated as variant under monitoring (VUM) and commonly referred to as 'pirola,' carries a newly gained *N*-glycosylation site at N354, which represents the first change in the RBD shielding since the ancestral strain.

The SARS-CoV-2 S RBD (aa 327–540) from the WHu-1 (China, 2019) to the EG.5.1 (China, 2023) shows two highly conserved *N*-glycosylation sites (*Harbison et al., 2022*), one at N331 and the other at N343 (*Watanabe et al., 2020a*). While glycosylation at N331 is located on a highly flexible region linking the RBD to the NTD, the N343 glycan covers a large portion of the RBD (*Casalino et al., 2020*; *Harbison et al., 2022*), stretching across the protein surface and forming a bridge connecting the two helical regions that frame the beta-sheet core, see *Figure 1b*. In this work, we show that removal of the N343 glycan induces a conformational change which in WHu-1, alpha, and beta allosterically controls the structure and dynamics of the RBM, see *Figure 1c*. In delta and omicron these effects are significantly dampened by mutations that strengthen the RBD architecture. Further to this molecular insight, we show that enzymatic removal of the N343 glycan affects binding of monosialylated ganglioside co-receptors (*Nguyen et al., 2022*) in the WHu-1 RBD, but not in delta. We also observe that the affinity of the RBD for GM1 $GM1_{os}$ and GM2 $GM1_{os}$ changes significantly across the VoCs, with beta and omicron exhibiting the weakest binding.

Ultimately, the molecular insight we provide in this work adds to the ever-growing evidence supporting the role of glycosylation in protein folding and structural stability. This information is not

only central to structural biology, but also critical to the design of novel COVID-19 vaccines that may or may not carry glycans (*Huang et al., 2022*), as well as instrumental to our understanding of the evolutionary mechanisms regulating the shield.

## Results

In this section, we start with a brief overview of the architecture of the RBD, we then explain how the RBD structure is modulated by interactions with ACE2 and why the N343 glycan is integral to its stability. We then describe how and why the loss of N343 glycosylation affects the RBD structure and its binding affinity for GM1$_{os}$ and GM2$_{os}$ to different degrees in the VoCs.

### SARS-CoV-2 S RBD structure and antigenicity

The SARS-CoV-2 S RBD encompasses both structured and intrinsically disordered regions. The structured region is supported by a largely hydrophobic beta sheet core, framed by two flanking, partially helical loops (aa 335–345 and aa 365–375), linked by a bridging *N*-glycan at N343, see *Figure 1b*. The aa 335–345 loop carries the N343 glycosylation site and it is part of an important antigenic region targeted by Class 2 and 3 antibodies (*Bangaru et al., 2022*; *Barnes et al., 2020*; *Carabelli et al., 2023*; *Chen et al., 2023*). In the bridging conformation, the N343 glycan pentasaccharide extends across the RBD beta sheet to reach the aa 365–375 loop forming highly populated hydrogen bonding and dispersion interactions with the backbone and with the sidechains of residues 365–375, see *Figure 1b,c* and *Appendix 1—figure 2*. The bridging N343 glycan shields the hydrophobic beta sheet core of the RBD from the surrounding water, preventing energetically unfavourable contacts. Due to its amphipathic nature, the N343 forms dispersion interactions with the hydrophobic residues of the beta sheet through its core GlcNAc-β(1-4)GlcNAc, while engaging in hydrogen bonds with the surrounding water and with the aa 365–375 helical loop. Notably, the key anchoring residues S371, S373, and S375 within this loop are all mutated to hydrophobic residues in all omicron variants (BA.1–2, BA.4–5, BQ.1.1, EG.5.1, XBB.1.5).

The RBM encompasses aa 439–506 and counts all the RBD residues in direct contact with ACE2 (*Lan et al., 2020*). The RBM is heavily targeted by both Class 1 and 2 antibodies (*Bangaru et al., 2022*; *Barnes et al., 2020*; *Carabelli et al., 2023*; *Chen et al., 2023*) and under high evolutionary pressure, with all VoCs carrying mutations in this region. As shown by earlier MD simulations studies (*Casalino et al., 2020*; *Harbison et al., 2022*; *Sztain et al., 2021*; *Williams et al., 2022*), the RBM in unbound S is largely unstructured and dynamic, an insight also supported by the low resolution cryo-EM maps of this region (*Gobeil et al., 2022*; *Walls et al., 2020*; *Wrapp et al., 2020*). The RBM's inherent flexibility is likely an important feature in the opening and closing mechanism of the RBD, where the N343 from adjacent RBDs engage with the protein in closed conformation and gate RBD opening (*Sztain et al., 2021*). The only relatively structured region of the RBM is what we define here on as the hydrophilic patch, see *Figure 1b and a* hairpin stabilised by a network of interlocking salt-bridges and polar residues, namely R454, R457, K458, K462, E465, D467, S469, and E471, that faces the interior of the S when the RBD is closed.

When in complex with ACE2 or with antibodies, the RBM adopts a structured fold, also shared by the SARS-CoV-1 S RBD (*Li et al., 2005*). In this conformation, only the terminal hairpin of the RBM (aa 476–486) retains a high degree of flexibility, as shown in this work and by others (*Williams et al., 2022*). The RBM-bound fold is stabilised by a hydrophobic patch supported by the stacking of the aromatic and aliphatic residues L455, F456, Y473, A475, see *Figure 1b*, which are part of the protein interface with ACE2. Notably, all residues in the hydrophobic and hydrophilic patches are highly conserved across the VoCs, possibly due to their critical function in inducing and/or stabilising the RBD into its ACE2-bound conformation. As an interesting observation, the loss of stacking in the hydrophobic patch due to the recent F456L mutation in the EG.5.1 variant (China, 2023) is recovered by the L455F mutation in the, appropriately named, FLip variant.

Based on evidence from screenings (*Bangaru et al., 2022*; *Carabelli et al., 2023*; *Chen et al., 2023*), we subdivided the RBD into three different antigenic regions known to be targeted by different classes of antibodies, see *Figure 1d*. Region 1 stretches from aa 337–353, which includes the N343 glycosylation site, and counts residues targeted by class 2 and 3 antibodies (*Bangaru et al., 2022*; *Carabelli et al., 2023*; *Harvey et al., 2021*). The aa sequence in Region 1 has been highly conserved

so far, allowing specific antibodies to retain their neutralisation activity across all VoCs, such as S309 (*Newby et al., 2023*; *Watanabe et al., 2020a*), whose binding mode also directly involves the N343 glycan (*Liu et al., 2021*). A notable and dramatic exception to this high degree of conservation in Region 1 is given by the BA.2.86 variant (Denmark, 2023), known as 'pirola,' where the K356T mutation introduces a new *N*-glycosylation sequon at N354. Region 2 coincides with the RBM, which, in addition to binding ACE2 and neutralising antibodies, used to bind the N370 glycan from adjacent RBDs (*Allen et al., 2023*; *Harbison et al., 2022*; *Watanabe et al., 2020b*). N370 glycosylation is lost in SARS-CoV-2 S with the RBM binding cleft available to bind glycan co-receptors, such as glycosaminoglycans (*Clausen et al., 2020*; *Kearns et al., 2022*), blood group antigens (*Nguyen et al., 2022*; *Wu et al., 2023*), monosialylated gangliosides (*Nguyen et al., 2022*), among others. Region 3 is a short, relatively structured loop stretching between aa 411–426, located on the opposite side of the RBD relative to Region 1, see *Figure 1d*.

## Effect of the loss of N343 glycosylation on the structure of the WHu-1, alpha, and beta RBDs

Results obtained for the WHu-1 strain and for the alpha (B.1.1.7) and beta (B.1.351) VoCs are discussed together due to their sequence and structure similarity, with alpha counting only one mutation (N501Y) and beta three mutations (K417N, E484K, and N501Y) relative to the WHu-1 RBD. Extensive sampling through conventional MD, i.e., 4 µs for the alpha and beta VoC and 8 µs with an additional 4 µs of Gaussian accelerated MD (GaMD) for the WHu-1 RBD, see *Appendix 1—table 1*, shows that the loss of N343 glycosylation induces a dramatic conformational change in the RBD, where one or both helical loops flanking the hydrophobic beta sheet core pull towards each other, see *Figure 2*. This conformational change can occur very rapidly upon removal of the *N*-glycan or after a longer delay due to the complexity of the conformational energy landscape. The data used for the analysis corresponds to systems that have reached structural stability, i.e., equilibrium; we discarded the timeframes corresponding to conformational transitions.

To explore the effects of the loss of N343 glycosylation in the WHu-1 RBD, we started the MD simulations from different conformations. In one set of conventional sampling MD trajectories (MD1) and in the GaMD simulations the starting structure corresponds to an open RBD from an MD equilibrated S ectodomain obtained in earlier work (*Harbison et al., 2022*). In this system, the RBM is unfolded and retains the maximum degree of flexibility. MD2 was started from a conformation corresponding to the ACE2-bound structure (*Lan et al., 2020*). Results obtained from MD2 and GaMD are entirely consistent with results from MD1, and thus are included as *Appendix 1—figure 1*. The GaMD simulation shows a lower degree of contact of the N343 glycan with the aa 365–375 stretch of the opposite loop, see *Figure 1c*, because most of the contacts are with residues further downstream from position 365. Nevertheless, the N343 remains engaged in a bridging conformation throughout the simulation. As shown by the RMSD values distributions, represented through Kernel Density Estimates (KDE) in *Figure 2*, the structure of Region 1 in the WHu-1 RBD is stable. In the glycosylated RBD, the stability of Region 1 is largely due to the contribution of the bridging N343 glycan, chosen here to be a complex biantennary FA2G2 form based on *Watanabe et al., 2020a*, forming hydrogen bonds with the residues in loop aa 365–375 throughout the simulations, see *Figure 1c* and *Appendix 1—figure 2*. Conversely, the conformation of the RBM (Region 2) is very flexible in both glycosylated and non-glycosylated forms. Loss of N343 glycosylation triggers a conformational change in the Region 1 of the WHu-1 RBD, shown by a broader KDE peak in *Figure 2a*. This conformational change ultimately triggers the complete detachment of the hydrophilic loop from Region 1, see *Figure 1c*, through rupture of the non-covalent interactions network between Y351 (Region 1) and S469 or T470 (Region 2) via of hydrogen bonding, and Y351 and L452 (Region 2) via CH-π stacking. Structural changes in Region 3 upon loss of glycosylation at N343 appear to be negligible.

The starting structure used for the simulations of the alpha RBD derives from the ACE2-bound conformation of the WHu-1 RBD (PDB 6M0J) modified with the N501Y mutation. The reconstructed glycan at N343 interacts with the aa 365–375 throughout the entire trajectory, but it adopts a stable conformation only after 830 ns, where we started collecting the data shown in *Figure 2b*. Again, we see that the loss of glycosylation at N343 causes a swift conformational change that brings the aa 335–345 and aa 365–375 loops closer together, see *Figure 2b*. This conformational change involves primarily Region 1, and just like the previous case, it ultimately determines the detachment of the

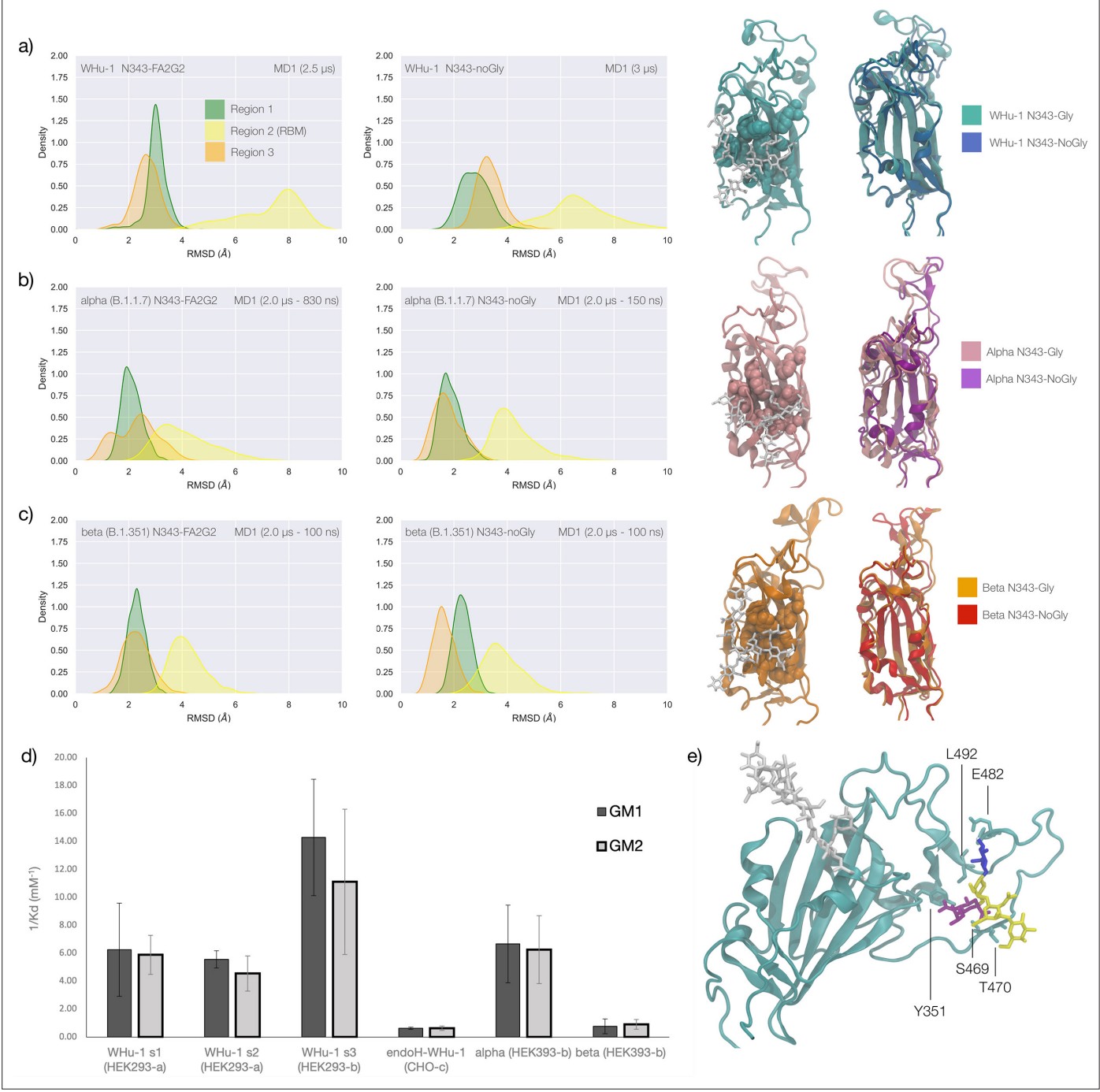

**Figure 2.** Conformational changes of the S receptor binding domain (RBD) structure in function of loss of N-glycosylation at N343 in WHu-1, alpha and beta variants and binding affinities for interactions between different RBDs and GM1/2os. (**a**) Kernel density estimates (KDE) plot of the backbone RMSD values calculated relative to frame 1 (t=0) of the trajectory for Region 1 (green) aa 337–353, Region 2 (yellow) aa 439–506, and Region 3 (orange) aa 411–426 of the glycosylated (left plot) and non-glycosylated (right plot) WHu-1 RBDs. Duration of the molecular dynamics (MD) sampling is indicated on the top-right corner of each plot with the conformational equilibration time subtracted as the corresponding data were not included in the analysis. Representative structures from the MD trajectories of the WHu-1 RBD glycosylated (cyan) and non-glycosylated (blue) at N343 are shown on the right-hand side of the panel. The N343 glycan (GlyTouCan-ID G00998NI) is rendered with sticks in white, the hydrophobic residues underneath the N343 glycan are highlighted with VDW spheres, while the protein structure is represented with cartoons. (**b**) KDE plot of the backbone RMSD values (see details in (**a**) above) calculated for the alpha (B.1.1.7) RBD glycosylated (left) and non-glycosylated (right) at N343. Representative structures from the MD simulation of the alpha RBDs are shown on the right-hand side of the panel, with the N343 glycosylated RBD shown with pink cartoons and the non-glycosylated alpha RBD in purple cartoons. (**c**) KDE plots of the backbone RMSD values calculated for the beta (B.1.351) RBD glycosylated (left) and

*Figure 2 continued on next page*

*Figure 2 continued*

non-glycosylated (right) at N343. Representative structures from the MD simulation of the beta RBDs are shown on the right-hand side of the panel, with the N343 glycosylated RBD shown with orange cartoons and the non-glycosylated alpha RBD in red cartoons. (**d**) Binding affinities (1/$K_d$, x$10^3$ M$^{-1}$) for interactions between different RBDs (including intact and endoF3 treated WHu-1 RBD and alpha and beta RBD) and the GM1$_{os}$ (GlyTouCan-ID G46613JI) and GM2$_{os}$ (GlyTouCan-ID G61168WC) oligosaccharides. HEK293a samples (*Nguyen et al., 2022*) and shown here as reference. HEK293b samples all carry FLAG and His tags and are shown for WHu-1 (glycosylated and treated with endoF3 treated), alpha, and beta sequences. Further details are in Appendix 1. (**e**) Predicted complex between the WHu-1 RBD and GM1$_{os}$, with GM1$_{os}$ represented with sticks in symbol nomenclature for glycans (SNFG) colours, the protein represented with cartoons (cyan), and the N343 with sticks (white). Residues directly involved in the GM1$_{os}$ binding or proximal are labelled and highlighted with sticks. All N343 glycosylated RBDs carry also a FA2G2 N-glycan (GlyTouCan-ID G00998NI) at N331, which is not shown for clarity. Rendering done with VMD (https://www.ks.uiuc.edu/Research/vmd/), KDE analysis with seaborn (https://seaborn.pydata.org/), and bar plot with MS Excel.

hydrophilic patch from the Y351 in Region 1. Also shown by the KDE plot in *Figure 2b and a* small conformational change in Region 3, which involves a partial disruption and refolding of a helical turn, can be observed during the trajectory of the N343 glycosylated alpha RBD. As in the previous case, the structure of Region 3 appears to be unaffected by N343 glycosylation, at least within the sampling accumulated in this work.

In the beta RBD (starting structure from PDB 7LYN) the reconstructed N343 glycan adopts a bridging conformation quite rapidly and retains this conformation throughout the trajectory with only minor deviations. The corresponding RMSD values KDE distributions for Regions 1–3, see *Figure 2c*, reflect this structural stability. The stability of the RBM (Region 2) is supported by interactions between Y351 (Region 1) and the hydrophilic loop, as noted earlier. Loss of glycosylation at N343 causes a rapid tightening of the RBD core helical loops towards each other, which again in this case ultimately causes the detachment of the hydrophilic loop from Y351 in Region 1 towards the end of the MD trajectory, i.e., after 1.9 µs of sampling.

## Effect of the loss of N343 glycosylation on the binding affinity of GM1$_{os}$ /2 $_{os}$ for the WHu-1, alpha, and beta RBDs

Earlier work from some of us shows that the WHu-1 RBD binds monosialylated gangliosides as co-receptors in SARS-CoV-2 infection (*Nguyen et al., 2022*). The 3D structure of the low affinity complex between the GM1$_{os}$ and the WHu-1 RBD has been proven difficult to obtain experimentally, yet we were able to obtain a promising model through a combination of multiple structural alignments of the GM1os to the WHu-1 RBD-bound N370 (*Harbison et al., 2022*) and extensive MD simulations, details in Appendix 1 and in *Newby et al., 2023*; *Watanabe et al., 2020a*. The predicted GM1$_{os}$ binding site is located at the junction between Region 1 and Region 2 of the WHu-1 RBD and it involves all the residues that stabilise the region, namely Y351, L452, S469, and T470, see *Figures 1c and 2e*. As part of our investigation of glycan co-receptors binding to the SARS-CoV-2 RBD, we used direct ESI-MS assay to determine the impact of the loss of N343 glycosylation on GM1$_{os}$ and GM2$_{os}$ binding. Here, we used endoF3-treatment to trim down the fucosylated biantennary and triantennary complex *N*-glycans into core nonfucosylated or fucosylated GlcNAc (Gn or GnF, respectively). LC-MS analysis suggests that *N*-glycans on N343 but not N331 of WHu-1 RBD were trimmed down (*Appendix 1—figure 6*). From the zero-charge mass spectra of endoF3-treated WT RBD (*Appendix 1—figure 5*), we performed glycan assignment (*Appendix 1—table 8*) and found that 31% of detected glycoforms contained Gn/GnF at N343, while the remaining was the intact form. Affinity data in *Figure 2d* show that the enzymatic removal of the N343 glycan from the WHu-1 RBD causes a complete loss of GM1$_{os}$/2$_{os}$ binding, which is consistent with both, the involvement of the junction between Regions 1 and 2 the binding and its allosteric control of the RBM dynamics. Furthermore, while binding of GM1$_{os}$ and GM2$_{os}$ to the alpha RBD appears to be slightly decreased relative to WHu-1, binding to the beta RBD is dramatically reduced. We can reconcile this finding with the mutation E484K in beta, which changes the key interaction between E484 and GM1$_{os}$, see *Figure 2* and with changes in structure and dynamics of the RBM terminal hairpin induced by mutations (*Williams et al., 2022*), which have also been suggested to affect the S opening kinetics (*Wang et al., 2021*).

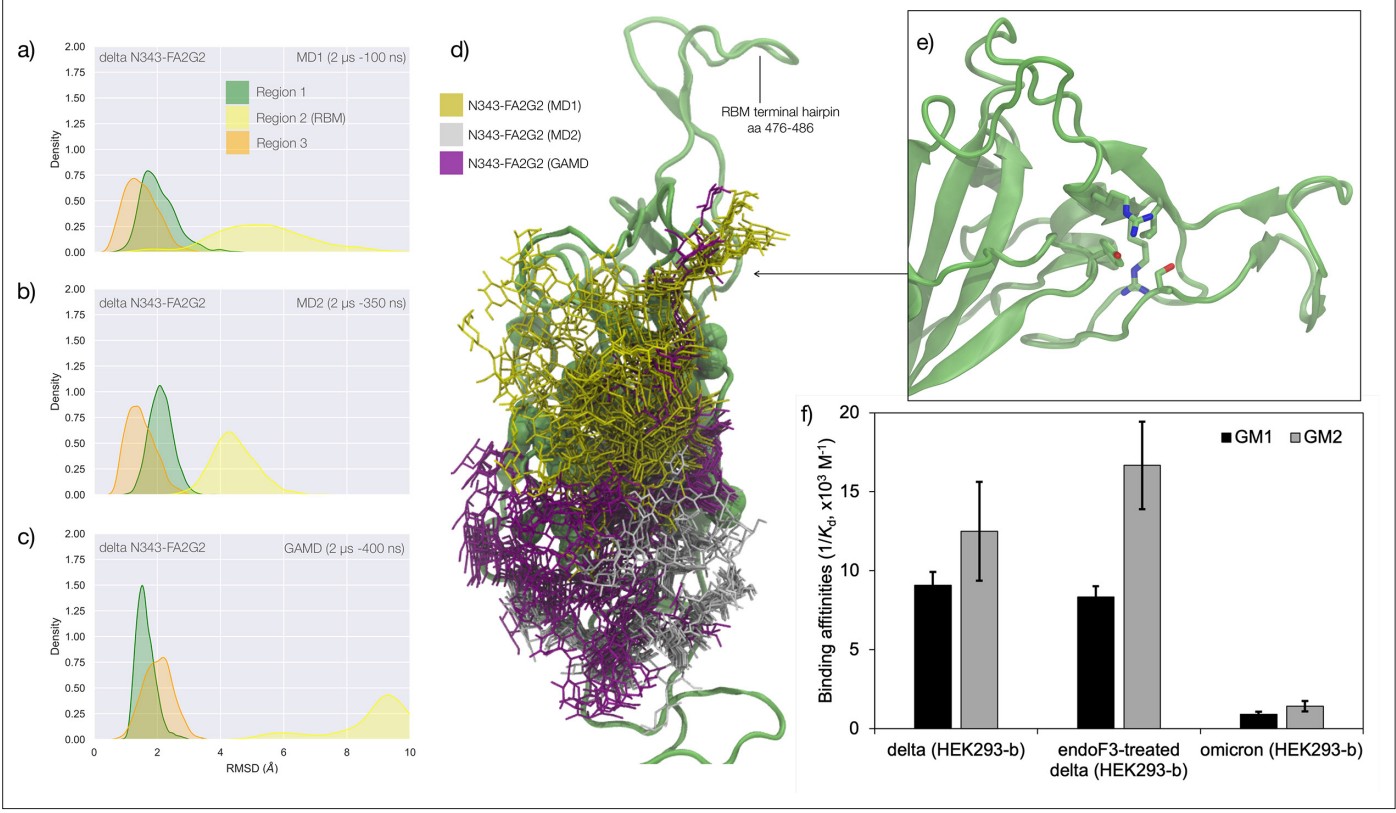

**Figure 3.** Conformational analysis and 3D structure of the S receptor binding domain (RBD) in the delta variant glycosylated at N343 with a diagram showing binding affinity to GM1/2os in the presence and absence of the glycan at N343 and in the omicron variant. (**a**) Kernel density estimates (KDE) plot of the backbone RMSD values calculated relative to frame 1 (t=0) of the MD trajectories (MD1) trajectory for Region 1 (green) aa 337–353, Region 2 (yellow) aa 439–506, and Region 3 (orange) aa 411–426 of the N343 glycosylated delta (B.1.617.2) RBD. The MD1 simulation was started from the open RBD conformation from the cryo-EM structure PDB 7V7Q. Based on the conformation of the N-glycan reconstructed at N353, the first 100 ns of the MD1 production trajectory were considered part of the conformational equilibration and not included in the data analysis. (**b**) KDE plot of the backbone RMSD values calculated relative to frame 1 (t=0) of the MD2 trajectory for Regions 1–3 (see details above) of the N343 glycosylated delta (B.1.617.2) RBD. The MD2 simulation was started from the open RBD conformation from the cryo-EM structure PDB 7V7Q with different velocities relative to MD1. The first 350 ns of the MD2 production trajectory were considered part of the conformational equilibration and not included in the data analysis. (**c**) KDE plot of the backbone RMSD values calculated relative to frame 1 (t=0) of the gaussian accelerated MD (GaMD) trajectory for Regions 1–3 of the N343 glycosylated delta (B.1.617.2) RBD. The first 400 ns of the GaMD production trajectory were considered part of the conformational equilibration and not included in the data analysis. (**e**) Graphical representation of the delta RBD with the protein structure (lime cartoon) from a representative snapshot from MD1. The N343 FA2G2 glycan (GlyTouCan-ID G00998NI) is represented in different colours, corresponding to the different molecular dynamics (MD) trajectories, as described in the legend, with snapshots taken at intervals of 100 ns. Residues in the hydrophobic core of the delta RBD are represented with VDW spheres partially visible under the *N*-glycans overlay. (**f**) Insert showing the junction between Regions 1 and 2 from the left-hand side of the RBD in (**e**). The residues involved in the network solidifying the junction are highlighted with sticks and labelled. (**f**) Affinities (1/K$_d$, x10$^3$ M$^{-1}$) for interactions between GM1$_{os}$ (GlyTouCan-ID G46613JI) and GM2$_{os}$ (GlyTouCan-ID G61168WC) oligosaccharides and the intact and endoF3-treated delta RBD and omicron RBD. Rendering done with VMD (https://www.ks.uiuc.edu/Research/vmd/), KDE analysis with seaborn (https://seaborn.pydata.org/), and bar plot with MS Excel.

## Effects of N343 glycosylation on the structure of the delta RBD

The delta (B.1.617.2) RBD carries two mutations, namely L452R and T478K, relative to the WHu-1 strain. The open RBD in the cryo-EM structure PDB 7V7Q was used as starting conformation for the MD simulations of both the glycosylated and the non-glycosylated delta RBDs. To understand how the mutations in delta affect the RBD structure and modulate the response to the loss of glycosylation at N343, we ran two uncorrelated conventional MD simulations (2 µs) and one GaMD simulation (2 µs) for both the glycosylated and non-glycosylated systems, for a total (cumulative) sampling of 12 µs. Results are shown in *Figure 3*. In the glycosylated delta RBD the N343 glycan is observed to be much more dynamic than in the WHu-1, alpha, and beta RBDs, engaging in contacts with different regions of the RBD in addition to the loop aa 365–375. In response to these fluctuations, the conformation of

the RBD remains stable with only minor deviations from the average structure of Regions 1 and 3. All trajectories, and in particular the results of the GaMD simulation in *Figure 3c*, show that RBM (Region 2) of delta is highly dynamic. This flexibility appears to involve specifically the terminal hairpin of the RBM (aa 476–486), which includes the T478K mutation, while the rest of the RBM is tightly anchored due to the L454R mutation. More specifically, the R452 of delta can establish a new hydrogen bond with Y351, in addition to S469 and T470, reinforcing the junction between Regions 1 and 2. Furthermore, in the delta RBM the role of L452 in CH-π stacking to Y351 is taken by the proximal L492, through a twist of the beta sheet, see *Figure 3e*. These interactions also contribute to reinforcing the R454 orientation, tightening the link with the RBM hydrophilic patch.

The effect of the loss of glycosylation at N343 on the delta RBD was assessed by running two uncorrelated MD simulations, one by conventional sampling (MD1 of 3 μs), and the other through enhanced sampling (GaMD of 2 μs). As a consequence of the L452R mutation shown in *Figure 3*, the tightening of the helical loops aa 335–345 and aa 365–375 over the hydrophobic core of the RBD occurring upon loss of glycosylation at N343 does not affect the structure and dynamic of the junction between Regions 1 and 2, see *Appendix 1—figure 3*. Results of the conventional MD simulation show that the tightening of the loops is mainly achieved by a larger displacement of the aa 365–375 loop rather than of Region 1, while the GaMD results show tightening of both loops, see *Appendix 1—figure 3*. In all simulations, the structure of the junction between Regions 1 and 2 remains undisturbed, with no detachment of the hydrophilic patch within the sampling we collected.

## Effect of the loss of N343 glycosylation on the binding affinity of GM1/2 for the delta RBD

To examine the effect of N343 glycosylation on glycan binding of delta RBD, we used the direct ESI-MS assay to quantify the binding affinities between endo F3-treated delta and GM1$_{os}$ and GM2$_{os}$. From the zero-charge mass spectra of endoF3-treated RBD, see *Appendix 1—figure 5*, we performed glycan assignment, see *Appendix 1—table 9*, and found that both N331 and N343 glycans were trimmed down to Gn/GnF. Direct ESI-MS data in *Figure 3f* show no loss of GM1/2 binding in the

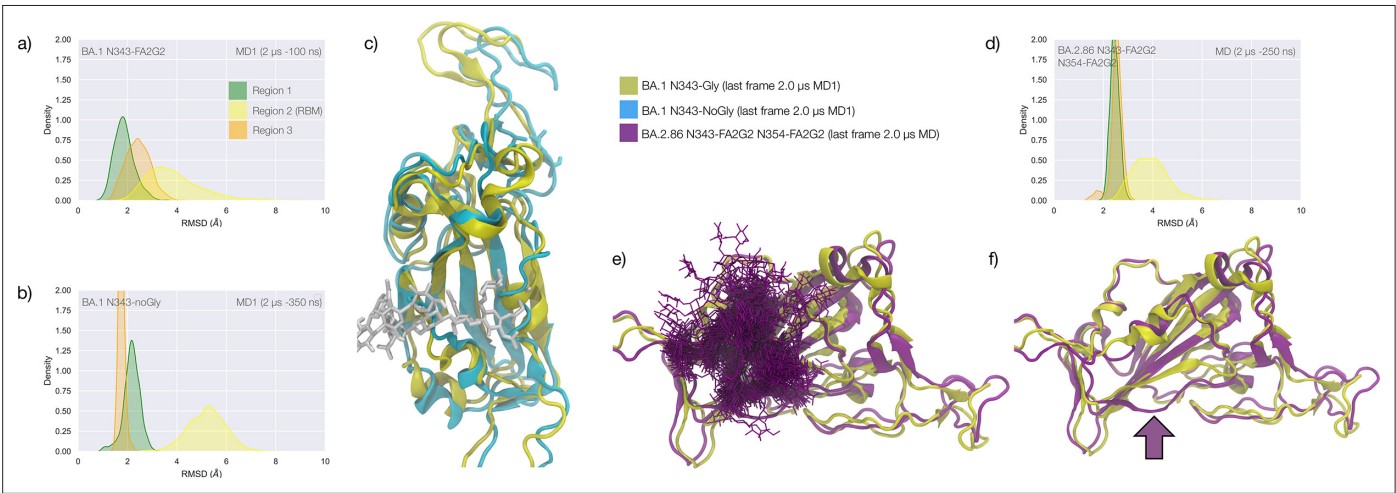

**Figure 4.** Conformational dynamics of the BA.1 and BA.2.86 S receptor binding domain (RBD) in function of N343 glycosylation. (**a**) Kernel density estimates (KDE) plot of the backbone RMSD values calculated relative to frame 1 (t=0) of the gaussian accelerated MD (GaMD) trajectory for Region 1 (green) aa 337–353, Region 2 (yellow) aa 439–506, and Region 3 (orange) aa 411–426 of the glycosylated omicron (BA.1) RBD. (**b**) KDE plot of the backbone RMSD values calculated relative to frame 1 (t=0) of the GaMD trajectory (see details above) of the non-glycosylated omicron (BA.1) RBD. (**c**) Graphical representation of the glycosylated (protein in yellow cartoons and N343-FA2G2 in white sticks, N331 omitted for clarity) and non-glycosylated (protein in cyan cartoons) of the omicron (BA.1) RBD. Structures correspond to the last frame of the GaMD trajectories, see details in the legend. (**d**) KDE plot of the backbone RMSD values calculated relative to frame 1 (t=0) of the molecular dynamics (MD) trajectory of the omicron BA.2.86 RBD glycosylated with FA2G2 N-glycans at N343, N354 and N331 (not shown). (**e**) Graphical representation of the omicron BA.2.86 RBD (protein in violet cartoons and N-glycans in violet sticks) structurally aligned to the glycosylated omicron (BA.1) RBD (protein in yellow cartoons) for reference. The N343 and N354 glycans are intertwined throughout the trajectory. (**f**) Same graphical representation of the omicron BA.2.86 and BA.1 RBDs with the N-glycans not shown. The purple arrow points to the displacement of the loop in response to the presence of the N354 glycan in BA.2.86. Rendering with VMD (https://www.ks.uiuc.edu/Research/vmd/) and KDE analysis with seaborn (https://seaborn.pydata.org/).

delta RBD upon loss of N343 glycosylation, which further supports the involvement of the Region 1–2 junction in sialylated glycans recognition.

## Effects of N343 glycosylation on the RBD structure in the omicron BA.1 SARS-CoV-2

The omicron BA.1 RBD carries 15 mutations relative to the WHu-1 strain, namely S371L, S373P, S375F, K417N, N440K, G446S, S477N, T478K, E484A, Q493R, G496S, Q498R, N501Y, Y505H, and T547K. The S371L, S373P, and S375F mutations, retained in all omicron VoCs including the most recently circulating XBB.1.5, EG.5.1, and BA.2.86, remove all hydroxyl sidechains that we have seen being involved in hydrogen bonding interactions with the N343 glycan in the WHu-1, alpha, beta, and delta RBDs, see *Figure 1c* and *Appendix 1—figure 3*. We investigated the effects of the loss of glycosylation at N343 in the structure of the BA.1 RBD through two sets of uncorrelated conventional MD simulations (MD1 and MD2) and one set of GaMD, with total cumulative simulation time of 12 µs. Starting structures correspond to the open RBD in PDB 7QO7 (MD1) and in PDB 7WVN (MD2 and GaMD), where the N343 glycan was reconstructed in different conformations, depending on the spatial orientation of the N343 sidechain. The results of the MD1 and GaMD simulations show that, despite the S371L, S373P, and S375F mutations, the N343 glycan is still forms stable contacts with the aa 365–375 loop, see *Figure 1c* and *Appendix 1—figure 3*, and these interactions contribute to the stability of the RBD structure, see *Figure 4*. In the starting structure, we used for MD2 the N343 glycan was built with the core pentasaccharide pointing away from the RBD hydrophobic core. Consequently, the *N*-glycan adopts different transient conformations during the MD2 trajectory, which terminate with an interaction with the hydrophobic interior of the RBD and with the N331 glycan, see *Appendix 1—figure 6*. In all simulations, the loss of glycosylation at N343 causes a tightening of the aa 335–345 and aa 365–375 loops, which in omicron is stabilised by more efficient packing of the aa 365–375 loop within the hydrophobic core, driven by the embedding of the L371 and F375 sidechains. The non-glycosylated RBD adopts a stable conformation where we do not see a detachment of the hydrophilic patch. The stabilising effect of the aa 365–375 loop mutations in omicron could not be tested by means of affinity for $GM1_{os}/2_{os}$ as omicron (BA.1) binds those epitopes only weakly, see *Figure 3f*. Based on the binding site we predicted by MD simulations, see *Figures 1d and 2e*, and as observed for beta, the loss of E484 due to the E484A mutation in omicron may negate $GM1_{os}/2_{os}$ binding.

The stability of the RBD structure is further enhanced by the presence of an additional glycosylation site at N354, which appeared in the recently detected omicron BA.2.86 'pirola' variant. As shown in *Figure 4d–f*, the *N*-glycans at N343 and N354 are tightly intertwined throughout the trajectory stabilising Region 1, also shielding the area very effectively. The presence of an additional *N*-glycosylation site at N354 also changes the conformation of the loop that hosts the site relative to the BA.1 starting structure we used as a template to run the MD simulation, see *Figure 4f*. To note, based on earlier glycoproteomics analysis (*Newby et al., 2023*; *Watanabe et al., 2020a*) and on the exposure to the solvent of the reconstructed glycan structure at N354, we chose to occupy all glycosylation sites with FA2G2 *N*-glycans.

## Discussion

Quantifying the role of glycosylation in protein folding and structural stability is a complex task due to the dynamic nature of the glycan structures (*Fadda, 2022*; *Woods, 2018*) and to the micro- and macro-heterogeneity in their protein functionalization (*Čaval et al., 2021*; *Riley et al., 2019*; *Struwe and Robinson, 2019*; *Thaysen-Andersen and Packer, 2012*; *Zacchi and Schulz, 2016*) that hinder characterization. Yet, the fact that protein folding occurs within a context where glycosylation types and occupancy can change on the fly, suggests that not all glycosylation sites are essential for the protein to achieve and retain a native fold and that those sites may be displaced without consequences to function. In this work, we investigated the structural role of the *N*-glycosylation at N343 in the SARS-CoV-2 S RBD, one of the most highly conserved sites in the viral phylogeny (*Harbison et al., 2022*). Extensive MD simulations in this and in earlier work by us and others (*Casalino et al., 2020*; *Grant et al., 2020*; *Harbison et al., 2022*; *Sikora et al., 2021*) show that the RBD core is efficiently shielded by this glycan. Furthermore, the N343 glycan has been shown to be mechanistically involved

in the opening and closing of the S (*Sztain et al., 2021*), making this glycosylation site functionally essential towards viral infection.

In this work, we performed over 45 µs of cumulative MD sampling with both conventional and enhanced schemes to show that the N343 glycan also plays a fundamental structural role in the WHu-1 SARS-CoV-2 and that this role has changed in the variants circulating thus far. While we cannot gauge how fundamental is N343 glycosylation towards correct folding, we see that the amphipathic nature of the complex *N*-glycan (*Watanabe et al., 2020a*) at N343 enhances the stability of the RBD architecture, bridging between the two partially helical loops that frame a highly hydrophobic beta sheet core. To note, we determined the same bridging structures also for oligomannose types *N*-glycans at N343 in earlier work (*Harbison et al., 2022*). In all variants, we observe that the removal of the glycan at N343 triggers a tightening of the loops in a response likely aimed at limiting access of water into the hydrophobic core. In WHu-1, alpha, and beta RBDs this event allosterically controls the dynamics of the RBM, ultimately causing the detachment of the hydrophilic patch and misfolding from the ACE2-recognized conformation. These results are in agreement with the drastic reduction of viral infectivity observed upon deletion of both N331 and N343 glycosylation in the WHu-1 strain (*Li et al., 2020*), where loss of structure may add to the loss of function through gating (*Sztain et al., 2021*) or vice versa.

As a functional assay to support this molecular insight, we determined how the binding affinity of the RBD for the oligosaccharides of the monosialylated gangliosides $GM1_{os}$ and $GM2_{os}$ is modulated by N343 glycosylation. These were shown in earlier work by us and others to function as co-receptors in WHu-1 infection (*Nguyen et al., 2022*). We predicted through extensive MD sampling that $GM1_{os}$ and $GM2_{os}$ bind the RBD into a site corresponding precisely to the location occupied by the 6-arm of an ancestral *N*-glycan at N370 (*Garozzo et al., 2022*; *Harbison et al., 2022*). Note, that the N370 site is still occupied by zoonotic sarbecoviruses (*Allen et al., 2023*). The $GM1_{os}$ binding site, see *Figure 2e*, is located precisely at the junction between Regions 1 and 2, which is disrupted by the loss of N343 glycosylation in WHu-1. Accordingly, we find that enzymatic removal of the N343 glycan abolishes $GM1_{os}$ and $GM2_{os}$ binding in the WHu-1 RBD, see *Figure 2d*. While we expect a similar loss of binding in alpha, within the context of a lower affinity relative to the WHu-1 RBD, we find that the beta RBD does not bind $GM1_{os}$ and $GM2_{os}$, regardless of its glycosylation state. Based on the structure of the GM1-RBD complex we identified, see *Figure 2e*, where E484 represents a key contact to the oligosaccharides, the mutation of E484K in beta may be key to the loss of binding, together with change in the RBM kinetics linked to this mutation and to variations within the same region (*Wang et al., 2021*; *Williams et al., 2022*).

In the delta variant, we observed that the L452R mutation is responsible for an increased structural stability of the RBD, reinforcing the non-covalent interactions network between Region 1 and the RBM. Indeed, the tightening of the loops occurring upon loss of N343 glycosylation does not trigger a misfolding of the RBM, see *Figure 3*. Accordingly, we observe that the delta RBD with the trimming down of N331 and N343 glycans shows no significant change in binding affinity for $GM1_{os}$ and $GM2_{os}$ relative to the fully glycosylated form, see *Figure 3f*.

In all omicron variants, including all the currently circulating VoCs and VUMs, the loop aa 365–375 that the N343 glycan hooks on, carries similar mutations, with the highly conserved S371, S373, and S375 all mutated to hydrophobic residues, see *Appendix 1—figure 1*. Our MD results on the BA.1 and BA.2.86 RBDs show that hydrophobic residues at positions 371, 372, and 373 can pack within the RBD core, while leading to a loop structure that can support the N343 glycan branches through interactions with the backbone, see *Figure 4*. We have shown for all variants that the contacts between the N343 glycan and the aa 365–375 stretch of the opposite loop are fairly equally distributed, between hydrophilic (hydrogen bonding) and hydrophobic (dispersion or van der Waals) type interactions, see *Figure 1c*. Therefore, it is expected that the loss of anchoring hydrogen bonding residues can be supported through other interactions. Within this context, the removal of the N343 glycan does still cause a tightening of the loops, yet through a different mechanism relative to the other variants that ultimately does not appear to affect the RBM dynamics. As in beta, for omicron, there is negligible binding of the N343 glycosylated RBD to $GM1_{os}$ and $GM2_{os}$, likely due to the E484A mutation, which based on the predicted structure of the GM1os/RBD complex, see *Figures 1d and 2e*, would deny a key interaction within the predicted binding site.

Taken together, our results show that since the WHu-1, alpha, and beta strains, the RBD has evolved to make the *N*-glycosylation site at N343 structurally dispensable. Within this framework, provided that an *N*-glycosylation site in the immediate vicinity of N343 is necessary for folding and for function, a shift of the site within the sequence can potentially occur. Such a modification may negatively affect recognition and binding by neutralising antibodies (*Liu et al., 2021*; *Piccoli et al., 2020*; *Pinto et al., 2020*) and thus promote evasion. We have also shown for the BA.2.86 that the new glycosylation site at N354 can effectively contribute to the stability of Region 1, while significantly increasing shielding.

Moreover, we show that specific VoCs lost affinity for monosialylated ganglioside oligosaccharides with a trend in agreement with a binding site located at the junction between Region 1 and the RBM, which is part of the N370 glycan binding cleft on the RBD (*Harbison et al., 2022*). This conclusion is further supported by how binding affinities for GM1$_{os}$ and GM2$_{os}$ change upon the loss of N343 glycosylation, in agreement with the MD results. Further to this, as mutations we identified dampened binding to monosialylated ganglioside oligosaccharides, it is also possible that further mutations may switch the affinity back on or determine a shift of preference of the RBD towards other glycans that can still be recognised within the N370 cleft. Further work is ongoing in this area.

Finally, the results from this work point to the importance of understanding the impact of *N*-glycosylation on protein structure and stability, with immediate consequences to COVID-19 vaccine design. Indeed, earlier work shows SARS-CoV-2 S-based protein vaccines with increased efficacy due to the removal of *N*-glycans (*Huang et al., 2022*), and of RBD-based vaccines in use and under development (*Cohen et al., 2022*; *Más-Bermejo et al., 2022*; *Valdes-Balbin et al., 2021*) that may be designed with and without *N*-glycans. The design of such constructs may benefit from understanding which *N*-glycosylation sites are structurally essential and which are dispensable. Further to this, our results show that taking into consideration the effects on *N*-glycosylation on protein structural stability and dynamics in the context of specific protein sequences, may be key to understanding epistatic interactions among RBD residues (*Rochman et al., 2022*; *Witte et al., 2023*), which would be otherwise difficult, where not impossible, to decipher.

# Materials and methods
## Computational methods

All simulations were performed using additive, all-atom force fields, namely the AMBER 14 SB parameter set (*Maier et al., 2015*) to represent protein atoms and counterions (200 mM of NaCl), GLYC-AM06j-1 (*Kirschner et al., 2008*) to represent glycans, and TIP3P for water molecules (*Jorgensen et al., 1983*). All production trajectories from conventional (deterministic) MD simulations were run for a minimum of 2 µs to ensure convergence. In some cases, we extended the simulations up to 3 µs to assess the stability of specific conformational transitions, where deemed necessary. All Gaussian accelerated MD (GaMD) (*Miao et al., 2015*; *Wang et al., 2021*) production trajectories were run for 2 µs. All simulations of the N343 glycosylated and non-glycosylated RBDs were started from identical 3D structures. The glycans at N331 and N343 were rebuilt as FA2G2 (GlyTouCan-ID G00998NI) based on glycoproteomics data (*Newby et al., 2023*; *Watanabe et al., 2020a*) with 3D structures from our GlycoShape database (*Ives et al., 2023*) (https://glycoshape.org). Further information on the RBD structures and PDB IDs for all variants, together with details on the MD systems set-up, equilibration protocols, and total sampling times allocations are available in Appendix 1. Sequences for all VoCs and VUM RBDs (aa 327–540) from https://viralzone.expasy.org/9556.

## Proteins and glycans

Expression and purification of recombinant WHu-1, Alpha, Beta, Delta, and Omicron RBD (EG[319]RVQP… VN[541]F, UniProt number P0DTC2) with C-terminal FLAG (SGDYKDDDDKG) and His tags (HHHHHHG) used in the current study were described elsewhere (*Akache et al., 2021*; *Colwill et al., 2022*). Mutations of SARS-CoV-2 RBD VOCs are shown in *Appendix 1—figure 1*. Proteins were purified using standard immobilised metal-ion affinity chromatography (IMAC), followed by size-exclusion chromatography on Superdex-75 to remove dimers as described (*Forest-Nault et al., 2022*). To obtain endo F3-treated WHu-1 and Delta RBD, 100 µg of each RBD was treated with endo F3 (purchased from New England Biolabs) in 1 x Glycobuffer (50 mM sodium acetate, pH 4.5) at 37 °C overnight. Each protein was dialyzed and concentrated against 100 mM ammonium acetate (pH 7.4) using an Amicon

0.5 mL microconcentrator (EMD Millipore) with a 10 kDa MW cutoff and stored at –80 °C until used. The concentrations of protein stock solutions were estimated by UV absorption (280 nM). The oligosaccharides of GM1 and GM2, Galβ1-3GalNAcβ1-4(Neu5Acα2–3)Galβ1-4Glc (MW 998.34 Da, GM1$_{os}$), and GalNAcβ1-4(Neu5Acα2–3)Galβ1-4Glc (MW 836.29 Da, GM2$_{os}$), respectively, were purchased from Elicityl SA (Crolles, France). 1 mM stock solutions of each glycan were prepared by dissolving a known mass of glycan in ultrafiltered Milli-Q water. All stock solutions were stored at –20 °C until needed.

## ESI-MS affinity measurements

Affinities ($K_d$) of glycan ligands for RBD were measured by the direct ESI-MS binding assay. The ESI-MS affinity measurements were performed in positive ion mode on a Q Exactive Orbitrap mass spectrometer (Thermo Fisher Scientific). The capillary temperature was 150 °C, and the S-lens RF level was 100; an automatic gain control target of $5 \times 10^5$ and a maximum injection time of 100 ms were used. The resolving power was 17,500. The instrument was equipped with a modified nanoflow ESI (nanoESI) source. NanoESI tips with an outer diameter (o.d.) of ~5 μm were pulled from borosilicate glass (1.2 mm o.d., 0.69 mm i.d., 10 cm length, Sutter Instruments, CA) with a P-97 micropipette puller (Sutter Instruments). A platinum wire was inserted into the nanoESI tip, making contact with the sample solution, and a voltage of 0.8 kV was applied. Each sample solution contained a given RBD (5 μM) and GM1$_{os}$ or GM2$_{os}$ (at three different concentrations ranging from 10 to 150 μM) in ammonium acetate (100 mM, pH 7.4). Data acquisition and pre-processing was performed using the Xcalibur software (version 4.1); ion abundances were extracted using the in-house software SWARM (*Kitov et al., 2019*). A brief description of the data analysis procedures used in this work is given as the Supporting Information.

## Protease digestion

20 μg of a given purified protein (intact and endoF3-treated WT RBDs) were dissolved in 100 μL of 8 M urea in 100 mM Tris-HCl (pH 8.0) containing 3 mM EDTA and incubated at room temperature for 1 hr. The denatured protein was then reduced with 5 μL of 500 mM dithiothreitol (DTT; Sigma-Aldrich) at room temperature for 1 hr; followed by alkylation with 12 μL of 500 mM iodoacetamide (Sigma-Aldrich) at room temperature for 20 min in the dark. The reaction was quenched by adding 5 μL of 250 mM DTT, and the solution buffer was exchanged using a 10 kDa Amicon Ultra centrifugal filter. The samples were loaded onto the filter and centrifuged at 14,000 × g for 15 min. The glycoprotein solution was subsequently digested with trypsin/chymotrypsin (substrate/enzyme (wt/wt) = 50) in 50 mM ammonium bicarbonate (pH 8.0) for 18 hr at 37 °C. The reaction was quenched by heat inactivation at 100 °C for 10 min. The lyophilized sample was stored at –20 °C until LC-MS analysis.

## Peptide analysis by reverse-phase liquid chromatography (RPLC)-MS/MS

The digested samples were separated using a RPLC-MS/MS on a Vanquish UHPLC system (Thermo Fisher Scientific) coupled with ESI-MS detector (Thermo Q Exactive Orbitrap). Peptide separation was achieved using a Waters Acquity UPLC Peptide BEH C18 column (1.7 μm, 2.1 mm × 150 mm; Waters). The eluents were 0.1% formic acid in water (solvent A) and 0.1% formic acid in acetonitrile (solvent B). The separation was performed at 60 °C. The following gradient was used for MS detection: t = 0 min, 95% solvent A (0.2 mL min$^{-1}$); t = 45 min, 40% solvent A (0.2 mL min$^{-1}$); t = 55 min, 5% solvent A (0.2 mL min$^{-1}$); t = 55.1 min, 95% solvent A (0.2 mL min$^{-1}$). During LC-MS analysis, the following parameters were used: sheath gas flow rate of 10 arbitrary units (AU), capillary temperature of 250 °C, and spray voltage of 1.5 kV. The mass spectra were acquired in positive mode with an m/z range of 200–3000 at a resolution of 70,000. The automatic gain control target was set at $1 \times 10^6$, and a maximum injection time of 100 ms was used. HCD mass spectra were acquired in the data-dependent mode for the five most abundant ions with a resolution of 17,500. Automatic gain control target, maximum injection time, and isolation window were set at $2 \times 10^5$, 200 ms, and 2.0 m/z, respectively. HCD-normalized collision energy was 25%. The data were recorded by Xcalibur (Thermo, version 4.1) and analyzed using Thermo BioPharma Finder software.

The peptide sequences (EG[319]RVQP…VN[541]FS with C-terminal FLAG (SGDYKDDDDKG) and His tags (HHHHHHG), UniProt number P0DTC2) were then identified using the theoretical digest feature of the software. Carbamidomethylation and carboxymethylation at cysteine residues were used as a

fixed modification. Common mammalian *N*- and *O*-glycans were also used as variable modifications. A precursor mass tolerance of 5 ppm was set. For quantification, the abundance of each *N*-glycan at each *N*-glycosylation site (N331 and N343) is the sum of MS areas under the peak curve divided by the corresponding charge states. Next, for each *N*-glycosylation site, the relative abundance of each *N*-glycan is calculated as its abundance over the total abundance of all *N*-glycans detected.

## Acknowledgements

The Science Foundation of Ireland (SFI) Frontiers for the Future Programme is gratefully acknowledged for financial support of CMI postdoctoral training (20/FFP-P/8809). The opinions, findings, and conclusions or recommendations expressed in this material are those of the author(s) and do not necessarily reflect the views of the Science Foundation Ireland. CMI and EF gratefully acknowledge ORACLE for Research for the generous allocation of computational and data storage resources. CAF acknowledges the Irish Research Council (IRC) for funding through the Government of Ireland Postgraduate Scholarship Programme (GOIPG/201912212). CMI, CAF, AMH, and EF acknowledge the Irish Centre for High-End Computing (ICHEC) for generous allocation of computational resources. Large part of the computational work described here was run on the HPC cluster *kay* at ICHEC, soon to be decommissioned. We would like to take this opportunity to thank *kay* for her invaluable service to the Irish scientific computing community, together with all the staff at ICHEC who took great care of her during the past 5 years. JSK and LN acknowledge the Natural Sciences and Engineering Research Council of Canada, the Canada Foundation for Innovation and the Alberta Innovation, and Advanced Education Research Capacity Program for funding. We are grateful to the members of the NRC-HHT Mammalian Cell Expression Section for their contribution to the cloning, expression, and purification of the various recombinant proteins used in this study and to the Pandemic Response Challenge Program of the National Research Council of Canada for its financial support.

## Additional information

### Funding

| Funder | Grant reference number | Author |
| --- | --- | --- |
| Science Foundation Ireland | 20/FFP-P/8809 | Callum M Ives |
| Irish Research Council | GOIPG/201912212 | Carl A Fogarty |
| Natural Sciences and Engineering Research Council of Canada | | Linh Nguyen |
| Canada Foundation for Innovation | | John Klassen |
| Alberta Innovation and Advanced Education Research Capacity Program | | John Klassen |

The funders had no role in study design, data collection and interpretation, or the decision to submit the work for publication.

### Author contributions

Callum M Ives, Linh Nguyen, Conceptualization, Data curation, Investigation, Writing – original draft; Carl A Fogarty, Aoife M Harbison, Investigation; Yves Durocher, Resources, Methodology; John Klassen, Conceptualization, Data curation, Formal analysis, Supervision, Writing – original draft, Project administration; Elisa Fadda, Conceptualization, Resources, Data curation, Formal analysis, Supervision, Funding acquisition, Investigation, Methodology, Writing – original draft, Project administration, Writing - review and editing

### Author ORCIDs
John Klassen ⓘ https://orcid.org/0000-0002-3389-7112

Elisa Fadda http://orcid.org/0000-0002-2898-7770

Reviewer #2 (Public Review): https://doi.org/10.7554/eLife.95708.3.sa1
Reviewer #3 (Public Review): https://doi.org/10.7554/eLife.95708.3.sa2
Author response https://doi.org/10.7554/eLife.95708.3.sa3

## Additional files

### Supplementary files
• MDAR checklist

### Data availability
All MD simulations data are available open access on https://doi.org/10.5281/zenodo.10441732.

The following dataset was generated:

| Author(s) | Year | Dataset title | Dataset URL | Database and Identifier |
|---|---|---|---|---|
| Fadda E | 2024 | MD simulations from "#GotGlycans: Role of N343 Glycosylation on the SARS-CoV-2 S RBD Structure and Co-Receptor Binding Across Variants of Concern | https://doi.org/10.5281/zenodo.10441732 | Zenodo, 10.5281/zenodo.10441732 |

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

# Appendix 1

## Materials and methods

### MD simulations

The RBD simulation systems were constructed from residues R327 to N540 of the SARS-CoV-2 S glycoprotein. The starting structure for WHu-1 MD1 was from the open RBD obtained from the simulation of the S ectodomain (*Casalino et al., 2020*), and the WHu-1 MD2 starting structure from PDB 6M0J. The alpha (B.1.1.7) starting structure was obtained from PDB 6M0J, with the N501Y mutation introduced with the mutagenesis tool in pymol (https://pymol.org/2/). The beta starting structures were from PDB 7LYN. All delta starting structures (MD1 and MD2, glycosylated and non-glycosylated) were obtained from PDB 7V7Q, with MD1 and MD2 productions started from different velocities. The omicron starting structures for MD1 was from PDB 7WVN, and the starting structure for MD2 from PDB 7QO7. Simulations of BA.2.86 were performed from PDB 7WVN, with sequence mutations introduced with the mutagenesis tool in pymol.

The charged N- and C-terminal residues were neutralised by capping with acetyl (ACE) and N-methylamide (NME) groups, respectively. The RBD was glycosylated with FA2G2 glycans at N331 and at N343. The structure of the (GlyTouCan-ID G00998NI) N-glycan was sourced from the GlycoShape glycan structure database (GDB), and was linked to the RBD using the Re-Glyco, a glycoprotein builder tool developed for GlycoShape (https://glycoshape.org). The systems were solvated in a water box with a minimum distance of 12 Å, and ions were added to neutralise any system charges to a total concentration of 200 mM NaCl.

All simulations were performed using AMBER18 (*Case et al., 2018*) on resources provided by the Irish Centre for High-End Computing (ICHEC). The AMBER 14 SB force field (*Maier et al., 2015*) was used to model proteins and ions, the GLYCAM06j-1 (*Kirschner et al., 2008*) force field was used to model glycans, and the TIP3P water model was used to model solvent molecules (*Jorgensen et al., 1983*).

The energy of the system was minimised in 500,000 steps using the steepest descent algorithm, with all heavy atoms of the protein and glycan restrained with a potential weight of 5 kcal.mol$^{-1}$.Å$^{-2}$. The system was then equilibrated in the NVT ensemble, with the system gradually heated from 0 to 100 K, and then from 100 K to 300 K. The system was then equilibrated in the NPT ensemble to maintain the pressure at 1 bar. Following this, most restraints were then removed, with restraints of a magnitude of 5 kcal.mol$^{-1}$.Å$^{-2}$ remaining in place on the caps, R327-N334, and G526-N540. This is because these regions of the RBD would be connected to the other domains of the spoke glycoprotein, and so would be structurally constrained in vivo. The system was then equilibrated for a further 100 ns, before production runs of ~2 µs, as summarised in *Appendix 1—table 1*.

The temperature was maintained at 300 K using Langevin dynamics with a collision frequency of 1 ps$^{-1}$, and the pressure was maintained at 1 bar using isotropic position scaling with a Berendsen barostat and a pressure relaxation time of 2 ps. Periodic boundary conditions were used throughout the simulations. The Van der Waals interactions were truncated at 11 Å and Particle Mesh Ewald (PME) was used to treat long-range electrostatics with B-spline interpolation of order 4. The SHAKE algorithm was used to constrain all bonds containing hydrogen atoms and to allow the use of a 2 fs time step for all simulations.

**Appendix 1—table 1.** Molecular dynamics (MD) sampling methods and corresponding times in the production phase of the simulations.

MD1 and MD2 indicate trajectories collected through conventional (deterministic) sampling. Gaussian accelerated MD (GaMD) indicates trajectories obtained through Gaussian accelerated MD sampling. Further details on the methodology and starting structures are in the text.

| Variant | N343-FA2G2 (µs) | | | N343 NoGly (µs) | | |
|---|---|---|---|---|---|---|
| | MD1 | MD2 | GaMD | MD1 | MD2 | GaMD |
| WHu-1 | 2.5 | 2.0 | 2.0 | 3.0 | 2.0 | 2.0 |
| Alpha (B.1.1.7) | 2.0 | - | - | - | 2.0 | - |
| Beta (B.1.351) | 2.0 | - | - | - | 2.0 | - |

*Appendix 1—table 1 Continued on next page*

*Appendix 1—table 1 Continued*

| Variant | N343-FA2G2 (µs) | | | N343 NoGly (µs) | | |
|---|---|---|---|---|---|---|
| | MD1 | MD2 | GaMD | MD1 | MD2 | GaMD |
| Delta (B.1.617.2) | 2.0 | 2.0 | 2.0 | 3.0 | - | 2.0 |
| Omicron (BA.1) | 2.0 | 2.0 | 2.0 | 2.0 | 2.0 | 2.0 |
| Omicron (BA.2.86) | 2.0 | - | - | - | - | - |
| Total (46.5) | 12.5 | 6.0 | 6.0 | 8.0 | 8.0 | 6.0 |

Additionally, Gaussian-accelerated MD (GaMD) simulations (*Miao et al., 2015*; *Wang et al., 2021*) were run using the GaMD module in AMBER18 (*Case et al., 2018*). GaMD simulations were performed for glycosylated and non-glycosylated RBD systems of the Whu-1, delta, and omicron variants. For these GaMD simulations, the starting structure for the Whu-1 variant was obtained from the simulation of the S ectodomain (*Casalino et al., 2020*), and the starting structures for the delta and omicron variants from PDBs 7V7Q and 7WVN, respectively. In brief, a 2 ns short conventional MD simulation was used to collect the required potential statistics for calculating GaMD acceleration parameters. This was followed by a 50 ns equilibration after adding the boost potential, and finally a 2 µs GaMD production simulation. The average and standard deviation of the system potential energies were calculated every 0.4 ns. GaMD simulations were run at the 'dual-boost' level by setting the reference energy to the lower bound. One boost potential was applied to the dihedral energetic term, and the other to the total potential energetic term. The upper limit of the boost potential standard deviation was set to 6 kcal.mol$^{-1}$ for both the dihedral and total potential energetic terms. The same temperature and pressure parameters were used in the conventional MD simulations.

## Structure of the RBD-GM1o Complex

The equilibrium structure of the unbound RBD was obtained from MD simulations of the SARS-CoV-2 spike (S) WHu-1 ectodomain (*Harbison et al., 2022*) and was screened for binding to the GM1 tetrasaccharide (GM1o) (*Nguyen et al., 2022*). The GM1o structure was built with the GLYCAM Carbohydrate Builder and equilibrated by MD simulations in bulk water separately from the RBD. The same GM1o structure is deposited in the GlycoShape GDB (https://glycoshape.org) where it can be downloaded. As docking did not produce any convincing binding poses, none of which proved to be stable when tested by MD simulations, we used the conformation of the N370 N-glycan bound across the RBD (*Harbison et al., 2022*) as a guideline for the generation of a set of potential RBD/GM1o complexes. The stability of all promising conformations produced this way was tested by extensive sampling through MD simulations, started with a restrained equilibration phase (100–500 ns), where the protein was equilibrated around the bound (fixed) GM1o, followed by unrestrained MD. The complex shown in *Figure 1d* was obtained after numerous binding and unbinding events that occurred during the unrestrained MD, and it remained stable for 700 ns. As an interesting note, the position of the Neu5Ac in the bound GM1o corresponds to the position that a terminal sialic acid in N370 would have when bound to the RBD.

## Assignment of glycan compositions of RBD glycoforms

For each RBD VOCs, the data were analysed using the measured molecular weights (MWs) of intact protonated WT, Alpha, Beta, Delta, and Omicron RBD (WTx, Ax, Bx, Dx, and Ox respectively, *Appendix 1—tables 2–6*). The MW of deglycosylated RBD was calculated based on the elemental composition corresponding to amino acid sequence (EG$^{319}$RVQP…VN$^{541}$FS, UniProt number P0DTC2) and any amino acid substitutions plus FLAG tag (SGDYKDDDDKG) and hexa histidine tag (HHHHHHG) and four disulfide bonds. The MWs of non-glycosylated RBD WT, Alpha, Beta, Delta, and Omicron are 27,352.6 Da, 27,401.6 Da, 27,386.6 Da, 27,422.7 Da, and 27,613.9 Da, respectively. Possible glycan compositions were simulated for the numbers of N-acetylhexosamines (N: HexNAc, N-acetylgalactosamine, and N-acetylglucosamine), hexoses (H: Hex, glucose, and galactose), fucoses (F: Fuc), and N-acetylneuraminic acids (S: Neu5Ac). Possible values of N, H, F, and S were calculated by considering reported *N*- and *O*-glycans. Possible MWs of RBD glycoforms were then calculated from the sum of aforementioned MW of RBD and MWs of glycan residues from each possible H_N_F_S combination.

## ESI-MS affinity measurements

The affinities of glycan ligands for RBD were measured by the direct ESI-MS binding assay (**Kitova et al., 2012**). For a monovalent protein-ligand (PL) interaction (**Equation S1**), the dissociation constant ($K_d$; **Equation S2**) can be calculated from the ratio ($R$) of total abundances of L-bound ($Ab$(PL)) to free P ions ($Ab$(P), **Equation S3**) measured by ESI-MS.

$$P + L \rightleftarrows PL \tag{S1}$$

$$K_d = \frac{[P]\,[L]}{[PL]} = \frac{[L]_0}{R} - \frac{[P]_0}{R+1} \tag{S2}$$

$$R = \frac{Ab\,(PL)}{Ab\,(P)} = \frac{[PL]}{[P]} \tag{S3}$$

where $[P]_0$ and $[L]_0$ are initial concentrations of P and L, respectively. The abundance ratio $R$ measured by ESI-MS is taken to be equal to the equilibrium concentration ratio in the solution. Because RBD consists of multiple species with distinct glycan compositions, and glycosylation can, in principle, influence binding, the affinities for L binding to individual RBD species (**Equation S4a** and **Equation S4b**) were determined.

$$P_1 + L \rightleftarrows P_1L \tag{S4a}$$

$$P_x + L \; P_xL \tag{S4b}$$

The corresponding equations of mass balance equations are,

$$[L]_0 = [L] + \sum_x [P_xL] \tag{S5a}$$

$$[P_1]_0 = [P_1] + [P_1L] \tag{S5b}$$

$$[P_2]_0 = [P_2] + [P_2L] \tag{S5c}$$

$$[P_x]_0 = [P_x] = [P_xL] \tag{S5d}$$

where $[P_x]_0$ is the initial concentration of a given $P_x$ species. Initial concentrations of individual RBD species were estimated from their relative abundances measured by ESI-MS, assuming uniform response factors (**Equation S6**).

$$[P_x]_0 = \frac{Ab\,(P_x)}{\sum_x Ab\,(P_x)}\,[P]_0 \tag{S6}$$

The affinity of a given $P_x$ species ($K_{dx}$) was calculated from **Equation S7**

$$K_{dx} = \frac{[P_x]\,[L]}{[P_xL]} = \frac{[L]_0}{R_x} - \frac{1}{R_x}\sum_x \frac{R_x\,[P_x]_0}{(1 + R_x)} \tag{S7}$$

where $R_x$ is the total abundance ratio of ligand-bound and free $P_x$ ions (**Equation S8**).

$$R_x = \frac{Ab\,(P_xL)}{Ab\,(P_x)} = \frac{[P_xL]_{eq}}{[P_x]_{eq}} \tag{S8}$$

In some instances, signal for one ligand-bound $P_x$ complex overlapped with signal for another (free) $P_x$ species. Spectral overlap was corrected for by considering the abundance ratio of the two $P_x$ species (for example $P_x$ and $P_{x+1}$) in the absence of L ($r$; **Equation S9**) and assuming that $P_x$ and $P_{x+1}$ exhibit identical affinities for L.

$$r = \frac{Ab\,(P_{x+1})}{Ab\,(P_x)} \tag{S9}$$

The corresponding $r$ value was then used to calculate the true $R_x$ value ($R_{x,corr}$; *Equation S10*) corrected for spectral overlap,

$$R_{x,cor} = \frac{Ab\left(P_xL\right)}{Ab\left(P_x\right)} - r$$

(S10)

which was used in *Equation S7* to calculate $K_{dx}$.

**Appendix 1—table 2.** Affinities of the oligosaccharides of GM1 and GM2 (GM1$_{os}$ and GM2$_{os}$, respectively) for WHu-1, Alpha, Beta, Delta, and Omicron receptor binding domain (RBD) and endoH-treated WHu-1 RBD and endoF3-treated WHu-1 and Delta RBD measured by ESI-MS for aqueous ammonium acetate (100 mM, pH 7.4) solutions containing a given RBD (5 µM) and glycan (three different initial concentrations ranging from 10 µM to 150 µM).

Data represent mean ± SD; n = 3 independent experiments for each glycan concentration.

| Variant | GM1$_{os}$ $K_d$ (mM) | GM2$_{os}$ $K_d$ (mM) |
|---|---|---|
| WHu-1 RBD (sample 1; HEK293-a) | 0.16 ± 0.04 | 0.17 ± 0.02 |
| WHu-1 RBD (sample 2; HEK293-a) | 0.18 ± 0.01 | 0.22 ± 0.03 |
| WHu-1 RBD (sample 3; HEK293-b) | 0.07 ± 0.01 | 0.09 ± 0.02 |
| endoF3-treated WHu-1 RBD (HEK293-b) | 3.6 ± 0.7 | 5.7 ± 0.6 |
| alpha (HEK293-b) | 0.15 ± 0.03 | 0.16 ± 0.03 |
| beta (HEK293-b) | 1.3 ± 0.4 | 1.10 ± 0.2 |
| delta (HEK293-b) | 0.11 ± 0.01 | 0.08 ± 0.02 |
| endoF3-treated delta (HEK293-b) | 0.12 ± 0.01 | 0.06 ± 0.01 |
| micron (HEK293-b) | 1.1 ± 0.2 | 0.70 ± 0.16 |

**Appendix 1—table 3.** Summary of molecular weights (MWs) of WHu-1 receptor binding domain (RBD) glycoforms (WTx) with respective relative abundances identified by ESI-MS and putative H_N_F_S combinations.

| RBD | Measured MW (Da) | Relative abundance | H_N_F_S combination | Theoretical MW (Da) | Mass difference (Da) |
|---|---|---|---|---|---|
| WT1 | 31342.4 | 25.5 | 9_11_2_0 | 31345.1 | 5.3 |
| WT2 | 31402.4 | 2.6 | 11_9_1_1 | 31408.0 | 2.4 |
| WT3 | 31505.2 | 42.7 | 9_9_0_3 | 31520.1 | 6.9 |
| WT4 | 31527.0 | 34.3 | 10_9_1_2 | 31537.1 | 2.1 |
| WT5 | 31609.4 | 19.0 | 11_10_3_0 | 31612.1 | 5.3 |
| WT6 | 31668.8 | 48.8 | 10_9_0_3 | 31682.1 | 5.4 |
| WT7 | 31690.2 | 56.2 | 11_9_1_2 | 31699.1 | 1.0 |
| WT8 | 31711.4 | 17.8 | 9_10_0_3 | 31723.2 | 3.7 |
| WT9 | 31771.1 | 43.9 | 12_10_3_0 | 31774.2 | 4.9 |
| WT10 | 31794.0 | 65.6 | 10_11_0_2 | 31797.2 | 4.8 |
| WT11 | 31816.5 | 26.5 | 10_9_1_3 | 31828.2 | 3.7 |
| WT12 | 31835.3 | 30.6 | 11_9_2_2 | 31845.2 | 1.9 |
| WT13 | 31854.4 | 44.6 | 9_10_1_3 | 31869.2 | 6.8 |
| WT14 | 31875.2 | 47.5 | 10_10_2_2 | 31886.2 | 3.1 |
| WT15 | 31896.4 | 35.8 | 11_10_1_2 | 31902.2 | 2.2 |

Appendix 1—table 3 Continued

| RBD | Measured MW (Da) | Relative abundance | H_N_F_S combination | Theoretical MW (Da) | Mass difference (Da) |
|---|---|---|---|---|---|
| WT16 | 31935.8 | 5.6 | 10_11_1_2 | 31943.3 | 0.6 |
| WT17 | 31957.8 | 85.0 | 11_11_0_2 | 31959.2 | 6.6 |
| WT18 | 31979.7 | 98.1 | 13_11_0_1 | 31992.3 | 4.6 |
| WT19 | 31001.2 | 61.3 | 9_10_2_3 | 32015.3 | 6.1 |
| WT20 | 32038.7 | 17.3 | 11_10_0_3 | 32047.3 | 0.6 |
| WT21 | 32060.1 | 55.7 | 12_10_1_2 | 32064.3 | 3.9 |
| WT22 | 32083.3 | 55.9 | 10_11_0_3 | 32088.3 | 3.0 |
| WT23 | 32105.5 | 1.1 | 12_11_0_2 | 32121.3 | 7.8 |
| WT24 | 32125.1 | 12.6 | 13_11_1_1 | 32138.3 | 5.2 |
| WT25 | 32144.3 | 72.6 | 10_12_1_2 | 32146.3 | 5.9 |
| WT26 | 32165.0 | 63.5 | 12_12_1_1 | 32179.3 | 6.4 |
| WT27 | 32185.7 | 39.8 | 11_10_1_3 | 32193.3 | 0.4 |
| WT28 | 32247.8 | 88.4 | 11_11_0_3 | 32250.3 | 5.5 |
| WT29 | 32270.3 | 100.0 | 13_11_0_2 | 32283.4 | 5.1 |
| WT30 | 32291.8 | 62.4 | 12_14_1_0 | 32294.4 | 5.4 |
| WT31 | 32348.8 | 58.7 | 12_10_1_3 | 32355.4 | 1.4 |
| WT32 | 32371.5 | 46.6 | 10_11_0_4 | 32379.4 | 0.1 |
| WT33 | 32434.4 | 69.7 | 10_12_1_3 | 32437.4 | 5.0 |
| WT34 | 32456.1 | 76.4 | 12_12_1_2 | 32470.4 | 6.3 |
| WT35 | 32477.0 | 43.6 | 11_10_1_4 | 32484.4 | 0.6 |
| WT36 | 32538.2 | 78.0 | 11_11_0_4 | 32541.4 | 4.8 |
| WT37 | 32560.9 | 79.6 | 13_11_0_3 | 32574.4 | 5.6 |
| WT38 | 32583.0 | 49.5 | 12_14_1_1 | 32585.5 | 5.5 |
| WT39 | 32615.2 | 19.1 | 11_10_0_5 | 32629.5 | 6.3 |
| WT40 | 32637.9 | 60.5 | 12_10_1_4 | 32646.5 | 0.6 |
| WT41 | 32659.6 | 39.0 | 10_11_0_5 | 32670.5 | 2.9 |
| WT42 | 32725.4 | 55.2 | 10_12_1_4 | 32728.5 | 4.9 |
| WT43 | 32747.7 | 58.2 | 12_12_1_3 | 32761.5 | 5.8 |
| WT44 | 32768.8 | 4.1 | 11_10_1_5 | 32775.5 | 1.3 |
| WT45 | 32826.8 | 1.7 | 11_11_0_5 | 32832.5 | 2.3 |
| WT46 | 32851.5 | 10.7 | 13_11_0_4 | 32865.5 | 6.1 |
| WT47 | 32873.7 | 4.3 | 12_14_1_2 | 32876.6 | 5.1 |
| WT48 | 32927.5 | 49.5 | 12_10_1_5 | 32937.6 | 2.1 |
| WT49 | 32949.8 | 15.6 | 10_11_0_6 | 32961.6 | 3.8 |
| WT50 | 33293.5 | 34.6 | 13_11_1_5 | 33302.7 | 1.2 |
| WT51 | 33583.8 | 9.4 | 13_11_1_6 | 33593.8 | 2.0 |

**Appendix 1—table 4.** Summary of molecular weights (MWs) of B.1.1.7 receptor binding domain (RBD) glycoforms (Alpha, Ax) with respective relative abundances identified by ESI-MS and putative H_N_F_S combinations.

| RBD | Measured MW (Da) | Relative abundance | H_N_F_S combination | Theoretical MW (Da) | Mass difference (Da) |
|---|---|---|---|---|---|
| A1 | 30894.6 | 6.4 | 9_10_0_0 | 30899.0 | 4.3 |
| A2 | 31080.7 | 17.1 | 9_8_2_1 | 31076.0 | 4.7 |
| A3 | 31137.8 | 5.3 | 9_9_3_0 | 31134.1 | 3.7 |
| A4 | 31160.2 | 3.2 | 9_7_2_2 | 31164.0 | 3.8 |
| A5 | 31184.9 | 3.7 | 9_10_0_1 | 31190.1 | 5.2 |
| A6 | 31243.1 | 13.0 | 10_8_2_1 | 31238.1 | 5.1 |
| A7 | 31265.9 | 10.9 | 10_11_0_0 | 31264.1 | 1.8 |
| A8 | 31300.5 | 4.5 | 10_9_3_0 | 31296.1 | 4.4 |
| A9 | 31319.6 | 3.4 | 8_10_2_1 | 31320.1 | 0.5 |
| A10 | 31346.6 | 2.6 | 10_10_0_1 | 31352.1 | 5.5 |
| A11 | 31369.5 | 19.4 | 11_10_1_0 | 31369.1 | 0.4 |
| A12 | 31405.1 | 26.4 | 8_9_2_2 | 31408.1 | 3.0 |
| A13 | 31427.3 | 13.6 | 11_11_0_0 | 31426.1 | 1.1 |
| A14 | 31449.4 | 4.9 | 7_10_4_1 | 31450.2 | 0.8 |
| A15 | 31482.0 | 4.6 | 9_10_2_1 | 31482.2 | 0.1 |
| A16 | 31505.7 | 3.4 | 7_11_3_1 | 31507.2 | 1.5 |
| A17 | 31529.1 | 30.9 | 12_10_1_0 | 31531.2 | 2.0 |
| A18 | 31553.5 | 35.3 | 10_11_0_1 | 31555.2 | 1.7 |
| A19 | 31590.8 | 27.8 | 12_11_0_0 | 31588.2 | 2.6 |
| A20 | 31611.6 | 16.5 | 8_10_2_2 | 31611.2 | 0.4 |
| A21 | 31658.9 | 9.8 | 11_10_3_0 | 31661.2 | 2.3 |
| A22 | 31693.9 | 8.8 | 13_10_1_0 | 31693.2 | 0.6 |
| A23 | 31715.2 | 96.9 | 11_11_0_1 | 31717.2 | 2.0 |
| A24 | 31737.6 | 13.7 | 12_11_1_0 | 31734.3 | 3.3 |
| A25 | 31773.3 | 19.8 | 9_10_2_2 | 31773.3 | 0.1 |
| A26 | 31793.9 | 13.4 | 12_12_0_0 | 31791.3 | 2.7 |
| A27 | 31819.9 | 31.4 | 12_10_1_1 | 31822.3 | 2.4 |
| A28 | 31844.4 | 20.4 | 10_11_0_2 | 31846.3 | 1.9 |
| A29 | 31880.1 | 13.3 | 12_11_0_1 | 31879.3 | 0.8 |
| A30 | 31901.1 | 78.6 | 8_10_4_2 | 31903.3 | 2.3 |
| A31 | 31920.8 | 4.8 | 11_12_0_1 | 31920.3 | 0.5 |
| A32 | 31937.3 | 2.7 | 12_12_1_0 | 31937.3 | 0.1 |
| A33 | 31957.6 | 10.8 | 13_12_0_0 | 31953.3 | 4.2 |
| A34 | 31978.1 | 10.0 | 9_11_2_2 | 31976.3 | 1.7 |
| A35 | 31006.1 | 100.0 | 11_11_0_2 | 31008.3 | 2.2 |
| A36 | 32027.7 | 5.2 | 12_11_3_0 | 32026.4 | 1.3 |
| A37 | 32061.8 | 4.9 | 9_10_2_3 | 32064.4 | 2.6 |

*Appendix 1—table 4 Continued on next page*

*Appendix 1—table 4 Continued*

| RBD | Measured MW (Da) | Relative abundance | H_N_F_S combination | Theoretical MW (Da) | Mass difference (Da) |
|---|---|---|---|---|---|
| A38 | 32081.4 | 53.5 | 12_12_0_1 | 32082.4 | 1.0 |
| A39 | 32109.6 | 10.5 | 12_10_1_2 | 32113.4 | 3.8 |
| A40 | 32137.5 | 13.8 | 10_11_2_2 | 32138.4 | 0.9 |
| A41 | 32191.8 | 82.8 | 13_11_3_0 | 32188.4 | 3.4 |
| A42 | 32212.7 | 3.2 | 11_12_0_2 | 32211.4 | 1.3 |
| A43 | 32227.5 | 2.6 | 12_12_1_1 | 32228.4 | 1.0 |
| A44 | 32266.5 | 39.1 | 9_11_2_3 | 32267.4 | 0.9 |
| A45 | 32297.9 | 48.2 | 11_11_0_3 | 32299.4 | 1.5 |
| A46 | 32372.0 | 53.5 | 12_12_0_2 | 32373.5 | 1.5 |
| A47 | 32393.9 | 5.0 | 11_10_4_2 | 32389.5 | 4.4 |
| A48 | 32446.4 | 31.2 | 11_11_1_3 | 32445.5 | 0.9 |
| A49 | 32483.1 | 35.2 | 13_11_3_1 | 32479.5 | 3.6 |
| A50 | 32504.3 | 4.7 | 11_12_2_2 | 32503.5 | 0.7 |
| A51 | 32557.2 | 43.9 | 14_12_3_0 | 32553.6 | 3.7 |
| A52 | 32590.0 | 10.0 | 11_11_2_3 | 32591.5 | 1.6 |
| A53 | 32631.8 | 25.2 | 10_12_2_3 | 32632.6 | 0.8 |
| A54 | 32663.5 | 34.6 | 12_12_0_3 | 32664.6 | 1.1 |
| A55 | 32707.1 | 4.5 | 11_13_2_2 | 32706.6 | 0.5 |
| A56 | 32737.2 | 44.7 | 11_11_1_4 | 32736.6 | 0.6 |
| A57 | 32774.4 | 6.2 | 13_11_3_2 | 32770.6 | 3.8 |
| A58 | 32811.6 | 21.6 | 12_12_1_3 | 32810.6 | 0.9 |
| A59 | 32848.2 | 24.6 | 14_12_3_1 | 32844.7 | 3.6 |
| A60 | 32869.8 | 4.8 | 12_13_2_2 | 32868.7 | 1.2 |
| A61 | 32887.8 | 3.5 | 13_13_1_2 | 32884.7 | 3.1 |
| A62 | 32922.7 | 34.7 | 10_12_2_4 | 32923.7 | 1.0 |
| A63 | 32954.7 | 11.5 | 12_12_0_4 | 32955.7 | 1.0 |
| A64 | 32998.0 | 16.2 | 11_13_2_3 | 32997.7 | 0.3 |
| A65 | 33028.4 | 28.5 | 11_11_1_5 | 33027.7 | 0.7 |
| A66 | 33050.4 | 3.3 | 12_11_4_3 | 33045.7 | 4.7 |
| A67 | 33072.5 | 4.0 | 12_14_2_2 | 33071.7 | 0.7 |
| A68 | 33102.6 | 30.2 | 12_12_1_4 | 33101.7 | 0.8 |
| A69 | 33138.5 | 8.1 | 14_12_3_2 | 33135.7 | 2.8 |
| A70 | 33176.8 | 16.5 | 13_13_1_3 | 33175.8 | 1.0 |
| A71 | 33213.2 | 22.7 | 15_13_3_1 | 33207.8 | 5.5 |
| A72 | 33250.3 | 3.8 | 12_12_2_4 | 33247.8 | 2.5 |
| A73 | 33288.0 | 23.1 | 16_14_3_0 | 33283.8 | 4.1 |
| A74 | 33320.0 | 12.6 | 11_11_3_5 | 33319.8 | 0.2 |
| A75 | 33362.3 | 13.4 | 12_14_2_3 | 33362.8 | 0.5 |

*Appendix 1—table 4 Continued on next page*

*Appendix 1—table 4 Continued*

| RBD | Measured MW (Da) | Relative abundance | H_N_F_S combination | Theoretical MW (Da) | Mass difference (Da) |
|---|---|---|---|---|---|
| A76 | 33393.8 | 21.4 | 12_12_1_5 | 33392.8 | 1.0 |
| A77 | 33467.9 | 24.3 | 13_13_1_4 | 33466.8 | 1.0 |
| A78 | 33504.4 | 8.7 | 15_13_3_2 | 33500.9 | 3.5 |
| A79 | 33542.5 | 11.6 | 14_14_1_3 | 33540.9 | 1.6 |
| A80 | 33578.2 | 15.7 | 16_14_3_1 | 33574.9 | 3.3 |
| A81 | 33613.2 | 4.6 | 13_13_0_5 | 33611.9 | 1.3 |
| A82 | 33653.0 | 18.2 | 12_14_2_4 | 33653.9 | 0.9 |
| A83 | 33685.1 | 8.2 | 14_14_2_3 | 33686.9 | 1.9 |
| A84 | 33727.9 | 11.9 | 11_13_3_5 | 33726.0 | 1.9 |
| A85 | 33759.2 | 17.4 | 13_13_1_5 | 33757.9 | 1.3 |
| A86 | 33833.4 | 16.8 | 14_14_1_4 | 33832.0 | 1.4 |
| A87 | 33868.9 | 7.3 | 16_14_1_3 | 33865.0 | 3.9 |
| A88 | 33907.0 | 11.8 | 13_13_0_6 | 33903.0 | 4.0 |
| A89 | 33944.0 | 12.4 | 13_13_0_6 | 33945.0 | 1.0 |
| A90 | 34018.6 | 14.8 | 12_14_2_5 | 34018.0 | 0.6 |
| A91 | 34051.1 | 10.3 | 13_13_3_5 | 34050.1 | 1.0 |
| A92 | 34125.0 | 11.1 | 14_14_1_5 | 34123.1 | 2.0 |
| A93 | 34198.1 | 9.5 | 13_13_0_7 | 34194.1 | 4.0 |
| A94 | 34309.0 | 7.9 | 13_15_0_6 | 34309.1 | 0.2 |
| A95 | 34599.3 | 3.5 | 13_15_2_6 | 34601.3 | 2.0 |
| A96 | 34823.3 | 2.2 | 16_13_1_7 | 34826.3 | 3.0 |
| A97 | 34886.3 | 8.7 | 16_14_2_6 | 34884.3 | 2.0 |

**Appendix 1—table 5.** Summary of molecular weights (MWs) of B.1.351 receptor binding domain (RBD) glycoforms (Beta, Bx) with respective relative abundances identified by ESI-MS and putative H_N_F_S combinations.

| RBD | Measured MW (Da) | Relative abundance | H_N_F_S combination | Theoretical MW (Da) | Mass difference (Da) |
|---|---|---|---|---|---|
| B1 | 31247.7 | 10.6 | 9_9_0_2 | 31255.0 | 7.3 |
| B2 | 31352.5 | 18.2 | 10_8_1_2 | 31360.0 | 7.6 |
| B3 | 31376.3 | 12.9 | 9_11_0_1 | 31370.0 | 6.2 |
| B4 | 31408.6 | 25.9 | 11_11_0_0 | 31403.1 | 5.5 |
| B5 | 31433.0 | 16.9 | 11_9_1_1 | 31434.0 | 1.0 |
| B6 | 31512.7 | 19.6 | 9_11_1_1 | 31516.1 | 3.4 |
| B7 | 31538.1 | 62.3 | 9_9_0_3 | 31546.1 | 8.0 |
| B8 | 31559.8 | 7.0 | 10_9_1_2 | 31563.1 | 3.2 |
| B9 | 31595.6 | 34.7 | 9_10_1_2 | 31604.1 | 8.6 |
| B10 | 31616.5 | 9.9 | 10_10_0_2 | 31620.1 | 3.6 |
| B11 | 31641.7 | 11.0 | 11_10_1_1 | 31637.1 | 4.6 |
| B12 | 31669.2 | 15.4 | 13_10_1_0 | 31670.1 | 0.9 |

*Appendix 1—table 5 Continued on next page*

*Appendix 1—table 5 Continued*

| RBD | Measured MW (Da) | Relative abundance | H_N_F_S combination | Theoretical MW (Da) | Mass difference (Da) |
|-----|------|------|------|------|------|
| B13 | 31700.2 | 100.0 | 10_9_0_3 | 31708.1 | 7.9 |
| B14 | 31723.8 | 64.4 | 11_9_1_2 | 31725.1 | 1.3 |
| B15 | 31758.7 | 10.9 | 10_10_1_2 | 31766.2 | 7.5 |
| B16 | 31778.5 | 16.2 | 11_10_0_2 | 31782.2 | 3.7 |
| B17 | 31803.0 | 12.5 | 12_10_1_1 | 31799.2 | 3.8 |
| B18 | 31829.5 | 42.9 | 10_11_2_1 | 31824.2 | 5.3 |
| B19 | 31886.1 | 84.9 | 12_9_1_2 | 31887.2 | 1.1 |
| B20 | 31906.0 | 32.1 | 10_10_0_3 | 31911.2 | 5.2 |
| B21 | 31962.0 | 22.6 | 10_11_1_2 | 31969.3 | 7.2 |
| B22 | 31991.3 | 58.7 | 10_9_2_3 | 31000.2 | 8.9 |
| B23 | 32014.8 | 31.8 | 11_9_1_3 | 32016.2 | 1.5 |
| B24 | 32066.4 | 53.9 | 11_10_0_3 | 32073.3 | 6.9 |
| B25 | 32089.4 | 36.3 | 12_10_1_2 | 32090.3 | 0.9 |
| B26 | 32142.2 | 10.0 | 12_11_0_2 | 32147.3 | 5.1 |
| B27 | 32177.0 | 45.9 | 12_9_1_3 | 32178.3 | 1.3 |
| B28 | 32196.7 | 21.8 | 10_10_0_4 | 32202.3 | 5.6 |
| B29 | 32251.5 | 48.1 | 13_10_1_2 | 32252.3 | 0.9 |
| B30 | 32271.1 | 10.1 | 11_11_0_3 | 32276.3 | 5.3 |
| B31 | 32326.4 | 14.9 | 14_11_1_1 | 32326.4 | 0.1 |
| B32 | 32356.4 | 33.3 | 11_10_0_4 | 32364.4 | 8.0 |
| B33 | 32379.5 | 27.6 | 12_10_1_3 | 32381.4 | 1.9 |
| B34 | 32431.3 | 48.8 | 12_11_0_3 | 32438.4 | 7.1 |
| B35 | 32454.7 | 29.5 | 11_9_2_4 | 32453.4 | 1.3 |
| B36 | 32541.6 | 24.4 | 13_10_1_3 | 32543.4 | 1.8 |
| B37 | 32560.9 | 20.8 | 11_11_0_4 | 32567.4 | 6.6 |
| B38 | 32617.1 | 43.1 | 14_11_1_2 | 32617.5 | 0.4 |
| B39 | 32637.1 | 9.6 | 12_12_0_3 | 32641.5 | 4.4 |
| B40 | 32722.1 | 35.2 | 12_11_0_4 | 32729.5 | 7.4 |
| B41 | 32744.8 | 17.2 | 13_11_1_3 | 32746.5 | 1.7 |
| B42 | 32796.8 | 34.1 | 11_10_1_5 | 32801.5 | 4.7 |
| B43 | 32820.3 | 13.6 | 12_10_2_4 | 32818.5 | 1.8 |
| B44 | 32927.0 | 9.0 | 12_12_0_4 | 32932.6 | 5.6 |
| B45 | 32983.0 | 30.0 | 13_10_2_4 | 32980.6 | 2.4 |
| B46 | 33109.9 | 10.9 | 12_10_2_5 | 33109.6 | 0.3 |
| B47 | 33162.3 | 26.9 | 12_11_1_5 | 33166.6 | 4.4 |
| B48 | 33273.5 | 21.2 | 13_10_2_5 | 33271.7 | 1.8 |
| B49 | 33348.1 | 27.5 | 14_11_2_4 | 33345.7 | 2.4 |
| B50 | 33452.7 | 24.3 | 12_11_1_6 | 33457.7 | 5.1 |

*Appendix 1—table 5 Continued on next page*

*Appendix 1—table 5 Continued*

| RBD | Measured MW (Da) | Relative abundance | H_N_F_S combination | Theoretical MW (Da) | Mass difference (Da) |
|---|---|---|---|---|---|
| B51 | 33528.0 | 25.5 | 13_12_1_5 | 33531.8 | 3.8 |
| B52 | 33637.9 | 17.3 | 14_11_2_5 | 33636.8 | 1.1 |
| B53 | 33713.1 | 20.7 | 13_10_3_6 | 33708.8 | 4.3 |
| B54 | 33817.9 | 21.0 | 13_12_1_6 | 33822.9 | 5.0 |
| B55 | 33893.0 | 17.0 | 14_13_1_5 | 33896.9 | 3.9 |
| B56 | 34003.3 | 13.8 | 13_10_3_7 | 33999.9 | 3.4 |
| B57 | 34183.0 | 10.0 | 14_13_1_6 | 34188.0 | 5.0 |
| B58 | 34548.7 | 7.7 | 15_14_1_6 | 34553.1 | 4.5 |
| B59 | 34693.1 | 13.5 | 15_14_2_6 | 34699.2 | 6.1 |

**Appendix 1—table 6.** Summary of molecular weights (MWs) of B.1.351.2 receptor binding domain (RBD) glycoforms (Delta, Dx) with respective relative abundances identified by ESI-MS and putative H_N_F_S combinations.

| RBD | Measured MW (Da) | Relative abundance | H_N_F_S combination | Theoretical MW (Da) | Mass difference (Da) |
|---|---|---|---|---|---|
| D1 | 31073.5 | 20.9 | 8_11_1_0 | 31079.0 | 5.5 |
| D2 | 31094.3 | 1.0 | 9_11_0_0 | 31095.0 | 0.7 |
| D3 | 31259.9 | 24.3 | 10_11_0_0 | 31257.1 | 2.8 |
| D4 | 31363.7 | 28.6 | 8_11_1_1 | 31370.1 | 6.4 |
| D5 | 31423.0 | 23.4 | 11_11_0_0 | 31419.1 | 3.9 |
| D6 | 31526.2 | 26.6 | 9_11_1_1 | 31532.2 | 5.9 |
| D7 | 31548.3 | 25.4 | 10_11_0_1 | 31548.2 | 0.2 |
| D8 | 31654.2 | 29.4 | 8_11_1_2 | 31661.2 | 7.0 |
| D9 | 31690.5 | 23.9 | 10_11_1_1 | 31694.2 | 3.7 |
| D10 | 31711.4 | 39.3 | 11_11_0_1 | 31710.2 | 1.2 |
| D11 | 31733.5 | 16.1 | 9_12_1_1 | 31735.2 | 1.7 |
| D12 | 31813.8 | 33.6 | 10_13_1_0 | 31809.3 | 4.5 |
| D13 | 31838.0 | 31.0 | 10_11_0_2 | 31839.3 | 1.3 |
| D14 | 31875.9 | 13.2 | 12_11_2_0 | 31873.3 | 2.6 |
| D15 | 31895.2 | 25.7 | 10_12_1_1 | 31897.3 | 2.1 |
| D16 | 31979.4 | 30.5 | 10_11_1_2 | 31985.3 | 5.9 |
| D17 | 31000.6 | 60.7 | 11_11_0_2 | 31001.3 | 0.8 |
| D18 | 32022.0 | 26.0 | 9_12_1_2 | 32026.3 | 4.3 |
| D19 | 32079.6 | 25.4 | 12_12_0_1 | 32075.3 | 4.2 |
| D20 | 32104.2 | 49.0 | 10_13_1_1 | 32100.4 | 3.8 |
| D21 | 32127.5 | 29.3 | 10_11_0_3 | 32130.3 | 2.8 |
| D22 | 32164.9 | 4.6 | 12_11_0_2 | 32163.4 | 1.6 |
| D23 | 32185.4 | 44.8 | 10_12_1_2 | 32188.4 | 3.0 |
| D24 | 32206.8 | 5.0 | 11_12_0_2 | 32204.4 | 2.4 |
| D25 | 32290.9 | 90.6 | 11_11_0_3 | 32292.4 | 1.5 |

*Appendix 1—table 6 Continued on next page*

*Appendix 1—table 6 Continued*

| RBD | Measured MW (Da) | Relative abundance | H_N_F_S combination | Theoretical MW (Da) | Mass difference (Da) |
|-----|------------------|--------------------|--------------------|--------------------|---------------------|
| D26 | 32312.4 | 32.4 | 9_12_1_3 | 32317.4 | 5.1 |
| D27 | 32348.4 | 26.0 | 11_12_1_2 | 32350.4 | 2.1 |
| D28 | 32369.1 | 34.1 | 12_12_0_2 | 32366.4 | 2.7 |
| D29 | 32394.4 | 45.5 | 10_13_1_2 | 32391.5 | 3.0 |
| D30 | 32475.9 | 50.0 | 10_12_1_3 | 32479.5 | 3.5 |
| D31 | 32496.5 | 22.2 | 11_12_0_3 | 32495.5 | 1.0 |
| D32 | 32553.6 | 0.7 | 11_13_1_2 | 32553.5 | 0.1 |
| D33 | 32581.7 | 100.0 | 11_11_0_4 | 32583.5 | 1.8 |
| D34 | 32603.7 | 34.7 | 12_11_3_2 | 32601.5 | 2.2 |
| D35 | 32638.2 | 0.3 | 11_12_1_3 | 32641.5 | 3.3 |
| D36 | 32658.4 | 41.3 | 12_12_0_3 | 32657.5 | 0.8 |
| D37 | 32683.3 | 21.8 | 10_13_1_3 | 32682.6 | 0.7 |
| D38 | 32766.4 | 41.9 | 10_12_1_4 | 32770.6 | 4.2 |
| D39 | 32843.0 | 8.0 | 11_13_1_3 | 32844.6 | 1.7 |
| D40 | 32873.4 | 68.5 | 11_11_0_5 | 32874.6 | 1.2 |
| D41 | 32895.2 | 18.5 | 12_11_3_3 | 32892.6 | 2.6 |
| D42 | 32948.2 | 45.1 | 12_12_0_4 | 32948.6 | 0.4 |
| D43 | 33238.9 | 42.5 | 12_12_0_5 | 33239.7 | 0.8 |
| D44 | 33313.9 | 30.2 | 13_13_0_4 | 33313.8 | 0.1 |
| D45 | 33604.6 | 37.1 | 13_13_0_5 | 33604.9 | 0.2 |
| D46 | 33896.1 | 29.5 | 13_13_0_6 | 33896.0 | 0.2 |
| D47 | 33969.9 | 9.2 | 12_12_3_6 | 33969.0 | 0.9 |
| D48 | 34186.4 | 21.6 | 13_13_0_7 | 34187.0 | 0.7 |
| D49 | 34260.3 | 9.0 | 14_14_0_6 | 34261.1 | 0.7 |

**Appendix 1—table 7.** Summary of molecular weights (MWs) of B.1.1.529 receptor binding domain (RBD) glycoforms (Omicron, Ox) with respective relative abundances identified by ESI-MS and putative H_N_F_S combinations.

| RBD | Measured MW (Da) | Relative abundance | H_N_F_S combination | Theoretical MW (Da) | Mass difference (Da) |
|-----|------------------|--------------------|--------------------|--------------------|---------------------|
| O1 | 32489.2 | 40.2 | 10_11_1_3 | 32487.6 | 1.5 |
| O2 | 32538.4 | 45.7 | 11_9_3_3 | 32535.7 | 2.7 |
| O3 | 32559.4 | 5.6 | 11_12_1_2 | 32561.7 | 2.3 |
| O4 | 32576.0 | 23.0 | 10_10_1_4 | 32575.7 | 0.3 |
| O5 | 32595.1 | 74.6 | 11_10_2_3 | 32592.7 | 2.4 |
| O6 | 32615.4 | 31.4 | 12_10_1_3 | 32608.7 | 6.7 |
| O7 | 32651.1 | 29.0 | 11_11_1_3 | 32649.7 | 1.4 |
| O8 | 32670.2 | 24.9 | 12_11_2_2 | 32666.7 | 3.4 |
| O9 | 32702.3 | 42.5 | 11_12_0_3 | 32706.7 | 4.4 |
| O10 | 32723.8 | 87.1 | 10_10_2_4 | 32721.7 | 2.1 |

*Appendix 1—table 7 Continued on next page*

*Appendix 1—table 7 Continued*

| RBD | Measured MW (Da) | Relative abundance | H_N_F_S combination | Theoretical MW (Da) | Mass difference (Da) |
|-----|------------------|--------------------|--------------------|--------------------|--------------------|
| O11 | 32744.0 | 2.5 | 11_10_3_3 | 32738.7 | 5.3 |
| O12 | 32761.0 | 11.3 | 12_10_2_3 | 32754.7 | 6.3 |
| O13 | 32779.8 | 53.6 | 10_11_1_4 | 32778.7 | 1.1 |
| O14 | 32798.7 | 23.0 | 11_11_2_3 | 32795.8 | 2.9 |
| O15 | 32829.1 | 62.1 | 11_9_3_4 | 32826.8 | 2.3 |
| O16 | 32852.5 | 27.6 | 11_12_1_3 | 32852.8 | 0.3 |
| O17 | 32885.7 | 85.7 | 11_10_2_4 | 32883.8 | 2.0 |
| O18 | 32905.7 | 29.7 | 12_10_3_3 | 32900.8 | 4.9 |
| O19 | 32942.2 | 12.1 | 11_11_1_4 | 32940.8 | 1.4 |
| O20 | 32960.8 | 40.8 | 12_11_2_3 | 32957.8 | 3.0 |
| O21 | 33014.9 | 100.0 | 12_12_1_3 | 33014.8 | 0.1 |
| O22 | 33035.6 | 25.5 | 11_10_3_4 | 33029.8 | 5.8 |
| O23 | 33070.2 | 52.1 | 12_13_0_3 | 33071.9 | 1.6 |
| O24 | 33089.8 | 52.7 | 11_11_2_4 | 33086.9 | 2.9 |
| O25 | 33120.4 | 52.4 | 13_11_2_3 | 33119.9 | 0.6 |
| O26 | 33144.7 | 28.3 | 11_12_1_4 | 33143.9 | 0.8 |
| O27 | 33177.5 | 50.7 | 11_10_2_5 | 33174.9 | 2.6 |
| O28 | 33195.9 | 33.2 | 12_10_3_4 | 33191.9 | 4.0 |
| O29 | 33217.4 | 13.2 | 12_13_1_3 | 33217.9 | 0.5 |
| O30 | 33233.9 | 1.6 | 11_11_1_5 | 33231.9 | 2.0 |
| O31 | 33251.6 | 58.4 | 12_11_2_4 | 33248.9 | 2.7 |
| O32 | 33271.2 | 15.4 | 13_11_3_3 | 33265.9 | 5.2 |
| O33 | 33306.1 | 83.2 | 12_12_1_4 | 33305.9 | 0.2 |
| O34 | 33326.7 | 28.9 | 13_12_2_3 | 33322.9 | 3.7 |
| O35 | 33361.5 | 30.3 | 12_13_0_4 | 33362.9 | 1.4 |
| O36 | 33380.8 | 61.4 | 13_13_1_3 | 33380.0 | 0.9 |
| O37 | 33402.5 | 0.3 | 14_13_0_3 | 33396.0 | 6.5 |
| O38 | 33435.9 | 27.5 | 11_12_1_5 | 33435.0 | 0.9 |
| O39 | 33454.8 | 15.1 | 12_12_2_4 | 33452.0 | 2.8 |
| O40 | 33486.2 | 24.5 | 14_12_0_4 | 33484.0 | 2.2 |
| O41 | 33508.3 | 18.3 | 12_13_1_4 | 33509.0 | 0.7 |
| O42 | 33542.1 | 37.0 | 12_11_2_5 | 33540.0 | 2.1 |
| O43 | 33561.5 | 24.8 | 15_13_0_3 | 33558.0 | 3.5 |
| O44 | 33597.5 | 25.2 | 12_12_1_5 | 33597.0 | 0.5 |
| O45 | 33617.3 | 33.1 | 13_12_2_4 | 33614.0 | 3.2 |
| O46 | 33671.8 | 47.2 | 13_13_1_4 | 33671.1 | 0.7 |
| O47 | 33691.7 | 16.4 | 14_13_0_4 | 33687.1 | 4.6 |
| O48 | 33726.3 | 13.7 | 13_14_0_4 | 33728.1 | 1.7 |

*Appendix 1—table 7 Continued on next page*

*Appendix 1—table 7 Continued*

| RBD | Measured MW (Da) | Relative abundance | H_N_F_S combination | Theoretical MW (Da) | Mass difference (Da) |
|---|---|---|---|---|---|
| O49 | 33745.7 | 35.3 | 12_12_2_5 | 33743.1 | 2.6 |
| O50 | 33776.2 | 16.1 | 14_12_2_4 | 33776.1 | 0.1 |
| O51 | 33800.4 | 17.0 | 12_13_1_5 | 33800.1 | 0.3 |
| O52 | 33833.8 | 3.0 | 14_13_1_4 | 33833.1 | 0.7 |
| O53 | 33851.7 | 28.4 | 15_13_0_4 | 33849.1 | 2.6 |
| O54 | 33874.5 | 10.3 | 13_14_1_4 | 33874.1 | 0.4 |
| O55 | 33907.8 | 21.7 | 13_12_2_5 | 33905.1 | 2.7 |
| O56 | 33927.2 | 14.4 | 14_12_3_4 | 33922.1 | 5.0 |
| O57 | 33963.1 | 27.1 | 13_13_1_5 | 33962.2 | 0.9 |
| O58 | 33982.3 | 13.7 | 14_13_0_5 | 33978.1 | 4.2 |
| O59 | 34037.0 | 32.6 | 14_14_1_4 | 34036.2 | 0.8 |
| O60 | 34058.2 | 3.7 | 13_12_3_5 | 34051.2 | 7.0 |
| O61 | 34090.9 | 8.8 | 12_13_1_6 | 34091.2 | 0.3 |
| O62 | 34110.9 | 11.6 | 13_13_2_5 | 34108.2 | 2.7 |
| O63 | 34142.7 | 22.5 | 15_13_0_5 | 34140.2 | 2.5 |
| O64 | 34165.6 | 5.1 | 13_14_1_5 | 34165.2 | 0.4 |
| O65 | 34199.3 | 6.6 | 15_14_1_4 | 34198.2 | 1.1 |
| O66 | 34217.1 | 20.9 | 14_12_1_6 | 34212.2 | 4.8 |
| O67 | 34274.9 | 20.8 | 14_13_2_5 | 34270.3 | 4.6 |
| O68 | 34327.5 | 26.7 | 14_14_1_5 | 34327.3 | 0.2 |
| O69 | 34346.6 | 15.5 | 13_12_3_6 | 34342.3 | 4.3 |
| O70 | 34384.5 | 0.9 | 13_10_4_7 | 34373.3 | 11.2 |
| O71 | 34402.4 | 23.3 | 13_13_2_6 | 34399.3 | 3.1 |
| O72 | 34457.1 | 9.0 | 13_14_1_6 | 34456.3 | 0.8 |
| O73 | 34506.7 | 18.8 | 14_12_1_7 | 34503.3 | 3.4 |
| O74 | 34529.7 | 3.1 | 12_13_4_6 | 34529.4 | 0.3 |
| O75 | 34564.3 | 25.0 | 14_13_0_7 | 34560.3 | 4.0 |
| O76 | 34620.0 | 26.6 | 14_14_1_6 | 34618.4 | 1.6 |
| O77 | 34638.4 | 23.0 | 15_14_2_5 | 34635.4 | 3.0 |
| O78 | 34658.8 | 12.7 | 16_14_3_4 | 34652.4 | 6.4 |
| O79 | 34768.4 | 0.9 | 14_14_2_6 | 34764.4 | 4.0 |
| O80 | 34798.6 | 29.8 | 14_12_1_8 | 34794.4 | 4.2 |
| O81 | 34840.4 | 0.5 | 13_13_3_7 | 34836.5 | 4.0 |
| O82 | 34858.9 | 20.4 | 14_13_0_8 | 34851.4 | 7.5 |
| O83 | 35019.8 | 14.5 | 15_13_0_8 | 35013.5 | 6.3 |
| O84 | 35041.2 | 10.5 | 13_14_3_7 | 35039.5 | 1.7 |
| O85 | 35220.7 | 23.8 | 15_14_0_8 | 35216.6 | 4.1 |
| O86 | 35312.3 | 2.6 | 15_13_2_8 | 35305.6 | 6.7 |
| O87 | 35354.0 | 19.8 | 15_14_1_8 | 35362.6 | 8.6 |

**Appendix 1—table 8.** Summary of molecular weights (MWs) of endoF3-treated WHu-Hu-1 receptor binding domain (RBD) glycoforms (eWTx) with respective relative abundances identified by ESI-MS and putative H_N_F_S combinations.

| RBD | Measured MW (Da) | Relative abundance | H_N_F_S combination | Theoretical MW (Da) | Mass difference (Da) | Number of trimmed N-glycans |
|---|---|---|---|---|---|---|
| eWT1 | 28116.9 | 4.5 | 1_3_0_0 | 28123.9 | 7.0 | 2 |
| eWT2 | 28243.9 | 26.4 | 3_2_0_0 | 28244.9 | 1.0 | 2 |
| eWT3 | 28357.3 | 14.5 | 1_2_1_1 | 28358.0 | 0.6 | 2 |
| eWT4 | 28618.0 | 23.8 | 1_4_2_0 | 28619.1 | 1.0 | 2 |
| eWT5 | 28881.1 | 2.4 | 3_3_1_1 | 28885.1 | 4.0 | 2 |
| eWT6 | 29029.2 | 18.7 | 3_3_2_1 | 29031.2 | 2.1 | 2 |
| eWT7 | 29143.8 | 1.1 | 6_4_0_0 | 29137.2 | 6.5 | 1 |
| eWT8 | 29318.2 | 1.1 | 5_5_1_0 | 29324.3 | 6.1 | 1 |
| eWT9 | 29406.5 | 8.2 | 5_4_1_1 | 29412.3 | 5.8 | 1 |
| eWT10 | 29454.5 | 14.1 | 4_5_1_1 | 29453.4 | 1.2 | 1 |
| eWT11 | 29649.4 | 0.6 | 7_5_1_0 | 29648.4 | 1.0 | 1 |
| eWT12 | 29736.3 | 9.6 | 5_7_1_0 | 29730.5 | 5.8 | 1 |
| eWT13 | 29753.5 | 1.3 | 5_5_0_2 | 29760.5 | 6.9 | 1 |
| eWT14 | 30084.9 | 1.3 | 7_5_2_1 | 30085.6 | 0.7 | 1 |
| eWT15 | 30103.4 | 10.3 | 5_6_1_2 | 30109.6 | 6.2 | 1 |
| eWT16 | 31033.0 | 27.6 | 8_6_2_3 | 31032.9 | 0.1 | 1 |
| eWT17 | 31216.4 | 17.2 | 7_7_1_4 | 31219.0 | 2.5 | 1 |
| eWT18 | 31416.8 | 8.0 | 7_8_1_4 | 31422.0 | 5.2 | 1 |
| eWT19 | 31437.4 | 0.9 | 8_8_0_4 | 31438.0 | 0.7 | 1 |
| eWT20 | 31482.3 | 9.0 | 7_9_2_3 | 31480.1 | 2.2 | 1 |
| eWT21 | 31505.7 | 38.4 | 9_9_0_3 | 31512.1 | 6.4 | 0 |
| eWT22 | 31543.3 | 41.0 | 9_7_1_4 | 31543.1 | 0.2 | 1 |
| eWT23 | 31564.8 | 55.2 | 7_8_2_4 | 31568.1 | 3.3 | 1 |
| eWT24 | 31607.4 | 2.2 | 11_10_3_0 | 31604.1 | 3.2 | 0 |
| eWT25 | 31644.1 | 27.3 | 8_9_2_3 | 31642.1 | 2.0 | 1 |
| eWT26 | 31668.2 | 82.8 | 10_9_0_3 | 31674.1 | 5.9 | 0 |
| eWT27 | 31691.0 | 33.8 | 11_9_1_2 | 31691.1 | 0.2 | 0 |
| eWT28 | 31708.4 | 9.9 | 9_10_0_3 | 31715.2 | 6.7 | 0 |
| eWT29 | 31727.6 | 58.6 | 10_10_1_2 | 31732.2 | 4.5 | 0 |
| eWT30 | 31748.3 | 42.7 | 11_10_0_2 | 31748.2 | 0.1 | 0 |
| eWT31 | 31770.9 | 29.2 | 12_10_3_0 | 31766.2 | 4.7 | 0 |
| eWT32 | 31795.9 | 25.7 | 10_11_0_2 | 31789.2 | 6.7 | 0 |
| eWT33 | 31833.6 | 35.0 | 11_9_2_2 | 31837.2 | 3.6 | 0 |
| eWT34 | 31854.4 | 95.9 | 9_10_1_3 | 31861.2 | 6.8 | 0 |
| eWT35 | 31872.6 | 14.1 | 10_10_2_2 | 31878.2 | 5.6 | 0 |
| eWT36 | 31893.5 | 1.5 | 11_10_1_2 | 31894.2 | 0.7 | 0 |
| eWT37 | 31912.9 | 51.6 | 12_10_0_2 | 31910.2 | 2.6 | 0 |
| eWT38 | 31933.2 | 26.1 | 10_11_1_2 | 31935.3 | 2.0 | 0 |

*Appendix 1—table 8 Continued on next page*

*Appendix 1—table 8 Continued*

| RBD | Measured MW (Da) | Relative abundance | H_N_F_S combination | Theoretical MW (Da) | Mass difference (Da) | Number of trimmed N-glycans |
|-----|------------------|--------------------|--------------------|--------------------|--------------------|------------------|
| eWT39 | 31958.1 | 100.0 | 11_11_0_2 | 31951.2 | 6.9 | 0 |
| eWT40 | 31980.4 | 28.5 | 13_11_0_1 | 31984.3 | 3.9 | 0 |
| eWT41 | 32017.7 | 36.8 | 10_10_1_3 | 32023.3 | 5.5 | 0 |
| eWT42 | 32060.9 | 6.2 | 12_10_1_2 | 32056.3 | 4.6 | 0 |

**Appendix 1—table 9.** Summary of molecular weights (MWs) of endoF3-treated B.1.351.2 receptor binding domain (RBD) glycoforms (eDx) with respective relative abundances identified by ESI-MS and putative H_N_F_S combinations.

| RBD | Measured MW (Da) | Relative abundance | H_N_F_S combination | Theoretical MW (Da) | Mass difference (Da) | Number of trimmed N-glycans |
|-----|------------------|--------------------|--------------------|--------------------|--------------------|------------------|
| eD1 | 28681.8 | 2.2 | 4_3_0_0 | 28680.1 | 1.7 | 2 |
| eD2 | 28720.3 | 24.9 | 3_4_0_0 | 28721.1 | 0.8 | 2 |
| eD3 | 28741.8 | 35.3 | 2_2_2_1 | 28736.1 | 5.6 | 2 |
| eD4 | 28762.6 | 9.4 | 2_5_0_0 | 28762.2 | 0.5 | 2 |
| eD5 | 28782.3 | 33.5 | 1_3_2_1 | 28777.2 | 5.2 | 2 |
| eD6 | 28803.6 | 25.9 | 3_3_2_0 | 28810.2 | 6.5 | 2 |
| eD7 | 28826.2 | 19.8 | 4_3_1_0 | 28826.2 | 0.1 | 2 |
| eD8 | 28867.3 | 82.4 | 3_4_1_0 | 28867.2 | 0.1 | 2 |
| eD9 | 28888.2 | 52.6 | 4_4_0_0 | 28883.2 | 5.0 | 2 |
| eD10 | 28907.6 | 20.6 | 3_2_2_1 | 28898.2 | 9.4 | 2 |
| eD11 | 28927.8 | 100.0 | 3_5_0_0 | 28924.2 | 3.6 | 2 |
| eD12 | 28949.2 | 57.5 | 3_3_1_1 | 28955.2 | 6.0 | 2 |
| eD13 | 28970.6 | 38.3 | 4_3_0_1 | 28971.2 | 0.6 | 2 |
| eD14 | 28990.7 | 30.3 | 2_4_1_1 | 28996.2 | 5.5 | 2 |
| eD15 | 29011.6 | 32.3 | 3_4_0_1 | 29012.2 | 0.6 | 2 |
| eD16 | 29032.7 | 24.1 | 4_4_1_0 | 29029.2 | 3.5 | 2 |
| eD17 | 29053.6 | 52.1 | 3_2_1_2 | 29043.2 | 10.4 | 2 |
| eD18 | 29074.7 | 51.9 | 3_5_1_0 | 29070.3 | 4.4 | 2 |
| eD19 | 29095.1 | 15.7 | 3_3_2_1 | 29101.3 | 6.2 | 2 |
| eD20 | 29114.2 | 37.7 | 4_3_1_1 | 29117.3 | 3.1 | 2 |
| eD21 | 29135.1 | 26.0 | 2_4_0_2 | 29141.3 | 6.2 | 2 |
| eD22 | 29158.3 | 34.9 | 3_4_1_1 | 29158.3 | 0.0 | 2 |
| eD23 | 29179.7 | 25.3 | 4_4_2_0 | 29175.3 | 4.4 | 2 |
| eD24 | 29198.8 | 17.6 | 3_2_2_2 | 29189.3 | 9.6 | 2 |
| eD25 | 29219.2 | 43.7 | 3_5_0_1 | 29215.3 | 3.9 | 2 |
| eD26 | 29240.2 | 24.3 | 3_3_1_2 | 29246.3 | 6.1 | 2 |
| eD27 | 29261.4 | 16.4 | 4_3_2_1 | 29263.3 | 1.9 | 2 |
| eD28 | 29281.5 | 6.4 | 2_4_1_2 | 29287.3 | 5.9 | 2 |
| eD29 | 29301.3 | 1.9 | 3_4_2_1 | 29304.3 | 3.0 | 2 |
| eD30 | 29302.0 | 1.2 | 3_4_0_2 | 29303.3 | 1.3 | 2 |
| eD31 | 29344.9 | 24.8 | 5_4_2_0 | 29337.4 | 7.6 | 2 |

*Appendix 1—table 9 Continued on next page*

*Appendix 1—table 9 Continued*

| RBD | Measured MW (Da) | Relative abundance | H_N_F_S combination | Theoretical MW (Da) | Mass difference (Da) | Number of trimmed N-glycans |
|-----|------------------|--------------------|--------------------|--------------------|---------------------|----------------------------|
| eD32 | 29366.8 | 5.0 | 3_5_1_1 | 29361.4 | 5.5 | 2 |
| eD33 | 29405.7 | 10.1 | 4_3_1_2 | 29408.4 | 2.7 | 2 |
| eD34 | 29427.1 | 3.8 | 2_4_2_2 | 29433.4 | 6.3 | 2 |

**Appendix 1—table 10.** Glycopeptide analysis of wild-type (WT) receptor binding domain (RBD).

| N-site | Peptide sequence | N-glycan | Retention time (min) | Measured m/z | Charge state | Mass accuracy (ppm) | MS area |
|--------|------------------|----------|---------------------|--------------|--------------|---------------------|---------|
| N331 | PNITNLCPFGEV | A2S2 | 18.0 | 1170.471 | 3 | −3.66 | 88472 |
| N331 | PNITNLCPFGEV | A2S2 | 18.0 | 1755.705 | 2 | −2.48 | 70771 |
| N331 | PNITNLCPFGEV | A1G1F | 18.2 | 1354.577 | 2 | 0.45 | 157724 |
| N331 | PNITNLCPFGEV | A1G0F | 18.3 | 1273.551 | 2 | 0.14 | 85745 |
| N331 | PNITNLCPFGEV | A1S1F | 18.6 | 1500.124 | 2 | 0.04 | 65793 |
| N331 | FPNIT | A2G2B | 27.8 | 1209.489 | 2 | −2.73 | 57370 |
| N343 | GEVFNATR | A2S1G1FB | 31.0 | 1579.132 | 2 | −3.43 | 44341 |
| N343 | NATRF | A2G2F | 6.3 | 1189.484 | 2 | 0.33 | 107145 |
| N343 | NATRF | A2G1F | 6.5 | 1107.955 | 2 | 0.30 | 46566 |
| N343 | NATRF | A2S1G1F | 7.0 | 1335.032 | 2 | 0.75 | 38217 |
| N343 | PFGEVFNATR | A2G2 | 13.5 | 1381.076 | 2 | −3.29 | 272764 |
| N343 | PFGEVFNATR | A2G1 | 13.6 | 1300.049 | 2 | −3.69 | 150239 |
| N343 | PFGEVFNATR | A2S1G1 | 13.7 | 1526.623 | 2 | −2.90 | 305875 |
| N343 | PFGEVFNATR | A2S1G0 | 13.8 | 1445.597 | 2 | −3.53 | 164686 |

**Appendix 1—table 11.** Glycopeptide analysis of endoF3-treated wild-type (WT) receptor binding domain (RBD).

| N-site | Peptide sequence | N-glycan | Retention time (min) | Measured m/z | Charge state | Mass accuracy (ppm) | MS area |
|--------|------------------|----------|---------------------|--------------|--------------|---------------------|---------|
| N331 | PNITNLCPFGEVF | A1G0F | 18.9 | 1347.585 | 2 | −3.88 | 21591 |
| N331 | PNITNLCPFGE | A2G0FB | 19.5 | 1427.087 | 2 | −5.81 | 83634 |
| N331 | PNITNLCPFGEV | A1G1F | 19.6 | 1354.575 | 2 | −1.09 | 112872 |
| N331 | PNITNLCPFGEV | A1G0F | 19.7 | 1273.551 | 2 | −1.21 | 52498 |
| N331 | PNITNLCPFGEV | A1S1F | 19.9 | 1500.122 | 2 | −1.51 | 59378 |
| N331 | PNITNLCPF | A2G1 | 26.2 | 827.350 | 3 | 1.33 | 49111 |
| N331 | PNITNLCPF | A2G1 | 26.9 | 1240.524 | 2 | 2.81 | 25854 |
| N331 | PNITNL | A1G1 | 28.6 | 964.915 | 2 | 0.19 | 19156 |
| N343 | NATRF | A2G2F | 6.3 | 1189.483 | 2 | 0.54 | 32460 |
| N343 | NATRF | A2G1F | 6.4 | 1107.955 | 2 | 0.19 | 47037 |
| N343 | NATRF | A2G0F | 6.5 | 1027.430 | 2 | 0.50 | 23831 |
| N343 | NATRF | A2S1G1F | 7.5 | 1335.032 | 2 | 0.75 | 34621 |
| N343 | NATRF | A2S2F | 7.8 | 1480.580 | 2 | 1.09 | 24974 |
| N343 | NATRF | GnF | 8.2 | 957.453 | 1 | 0.28 | 17273 |
| N343 | NATRF | GnF | 8.2 | 479.230 | 2 | 0.53 | 255747 |
| N343 | NATR | A2G0 | 18.0 | 880.363 | 2 | −2.3 | 34690 |

*Appendix 1—table 11 Continued on next page*

*Appendix 1—table 11 Continued*

| N-site | Peptide sequence | N-glycan | Retention time (min) | Measured m/z | Charge state | Mass accuracy (ppm) | MS area |
|--------|------------------|----------|---------------------|--------------|--------------|---------------------|---------|
| N343 | NATRFASVY | A2S1G1 | 9.4 | 736.560 | 4 | 4.72 | 103643 |
| N343 | GEVFNATR | GnF | 11.3 | 621.797 | 2 | 0.83 | 14586 |
| N343 | GEVFNATR | GnF | 15.5 | 414.868 | 3 | 3.39 | 15706 |
| N343 | GEVFNATR | GnF | 15.7 | 414.868 | 3 | 3.78 | 18198 |
| N343 | GEVFNATRF | GnF | 17.6 | 695.331 | 2 | 1.23 | 17392 |
| N343 | GEVFNATRF | A2G0F | 23.2 | 1243.537 | 2 | 4.76 | 17779 |

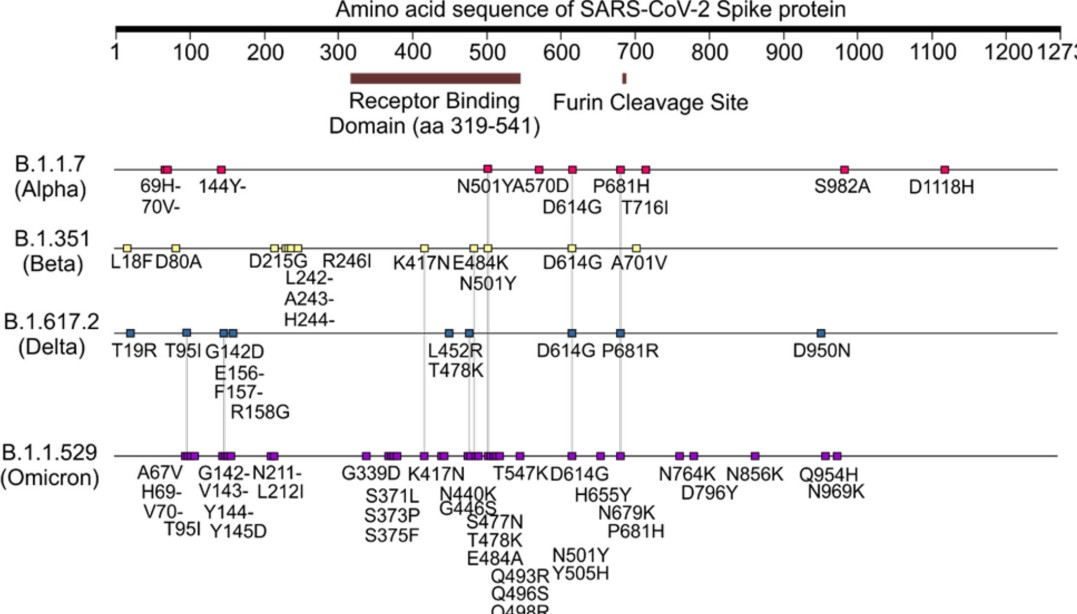

**Appendix 1—figure 1.** Amino acid sequence of SARS-CoV-2 spike (S) protein variants and mutations on receptor binding domain (RBD) used in this study.

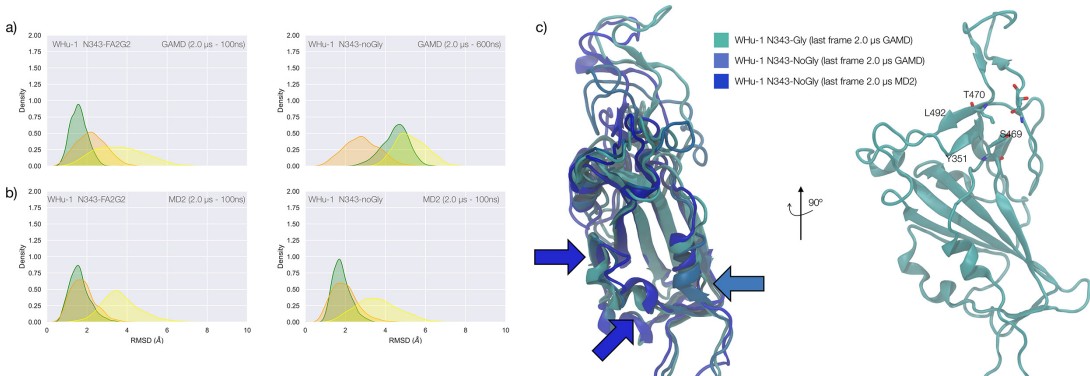

**Appendix 1—figure 2.** Conformational dynamics of the WHu-1 receptor binding domain (RBD) in function of N343 glycosylation. (**a**) Kernel density estimates (KDE) plots of the backbone RMSD values calculated relative to frame 1 (t=0) of the gaussian accelerated MD (GaMD) trajectory for Region 1 (green) aa 337–353, Region 2 (yellow) aa 439–506, and Region 3 (orange) aa 411–426 of the N343 glycosylated and non-glycosylated WHu-1 RBD. The GaMD simulations were started from the structure of the RBD in PDB 6M0J. The first 100 ns of the N343 glycosylated RBD trajectory were considered part of the conformational equilibration and not included in the data analysis. The first 600 ns of the trajectory obtained for the N343 non-glycosylated WHu-1 RBD were considered
*Appendix 1—figure 2 continued on next page*

*Appendix 1—figure 2 continued*

part of the conformational equilibration and not included in the data analysis. (**b**) KDE plots of the backbone RMSD values calculated relative to frame 1 (t=0) of the conventional molecular dynamics (MD) trajectory MD2 (see details above). MD2 was started from the conformation of the RBD in PDB 6M0J. The first 100 ns of both trajectories were considered part of the conformational equilibration and not included in the data analysis. (**c**) Graphical representation of the structural alignment of N343 glycosylated and non-glycosylated RBDs from the last frames of the GaMD and MD2 trajectories. Colour-coded arrows (see legend) indicate where the main conformational changes leading to the tightening of the helices occur in each system. Proteins are represented by cartoons and N343 glycan is not represented for clarity. On the right-hand side a rotation of the RBD by 90°Clockwise shows how the hydrophilic loop in Region 2 is still well connected to Y351 in Region 1 at the end of the GaMD N343 glycosylated WHu-1 RBD. Rendering done with VMD (https://www.ks.uiuc.edu/Research/vmd/) and KDE analysis with seaborn (https://seaborn.pydata.org/).

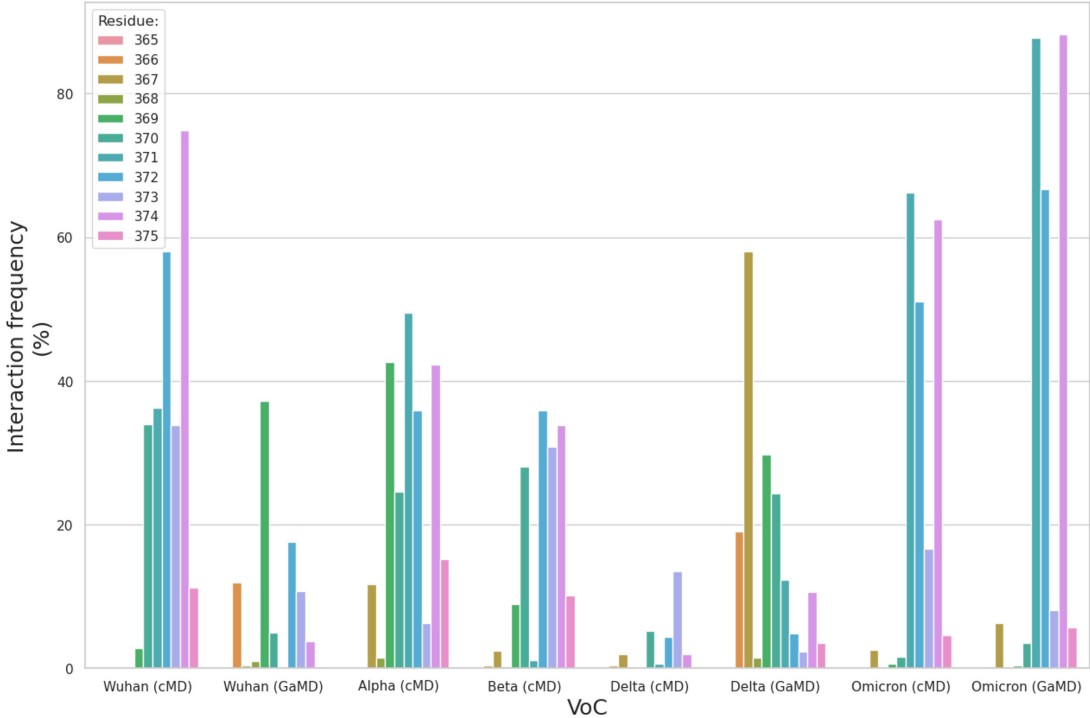

**Appendix 1—figure 3.** Bar plot of the interaction frequencies (%) of the N343 *N*-glycan with the different residues within the aa 365–375 loop for each variants of concern (VoC). The interactions include both hydrogen bonding and dispersion (van der Waals) contacts.

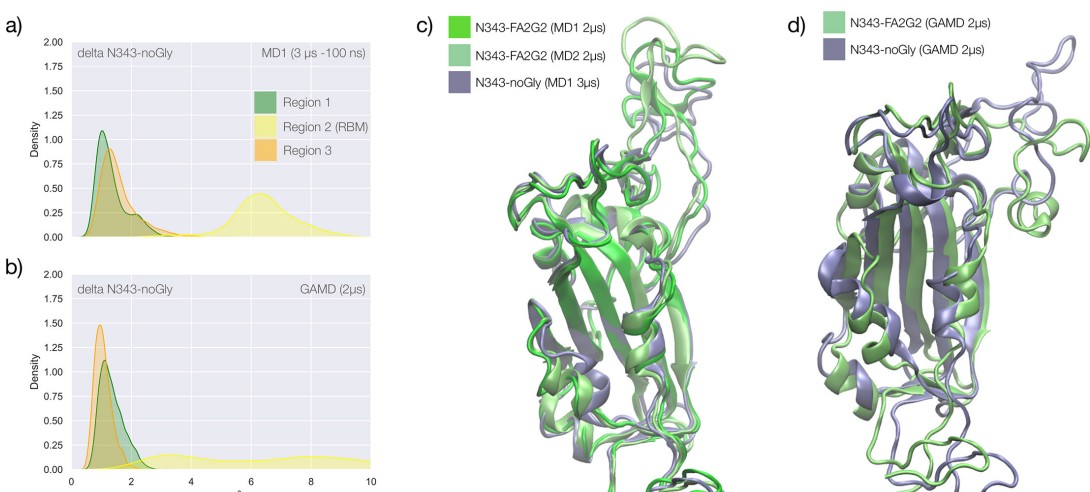

**Appendix 1—figure 4.** Conformational dynamics of the delta receptor binding domain (RBD) in function of N343 glycosylation. (**a**) Kernel density estimates (KDE) plot of the backbone RMSD values calculated relative to frame 1 (t=0) of the MD1 trajectory for Region 1 (green) aa 337–353, Region 2 (yellow) aa 439–506, and Region 3 (orange) aa 411–426 of the N343 non-glycosylated delta RBD. (**a**) KDE plot of the backbone RMSD values calculated relative to frame 1 (t=0) of the gaussian accelerated MD (GaMD) trajectory of the N343 non-glycosylated delta RBD. (**c**) Graphical representations of the delta RBD structures from the last frame of the conventional simulation MD1 (N343 glycosylated and non-glycosylated) and MD2 (N343 glycosylated) with colourings indicated in the legend. Protein is represented with cartoons and the N343 and N331 glycans are omitted for clarity. (**c**) Graphical representations of the structurally aligned delta RBD structures from the last frame of the accelerated simulation GaMD (N343 glycosylated and non-glycosylated). Protein is represented with cartoons and the N343 and N331 glycans are omitted for clarity. Rendering done with VMD (https://www.ks.uiuc.edu/Research/vmd/) and KDE analysis with seaborn (https://seaborn.pydata.org/).

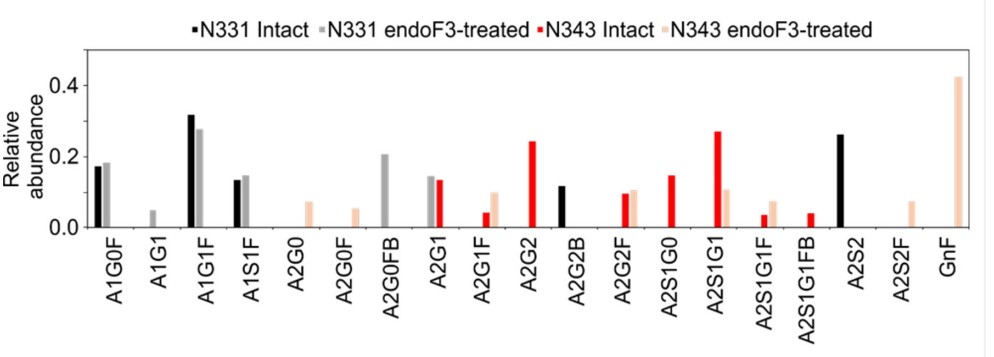

**Appendix 1—figure 5.** Relative abundance of *N*-glycans at N331 and N343 on the wild-type (WT) receptor binding domain (RBD) before and after endoF3 treatment.

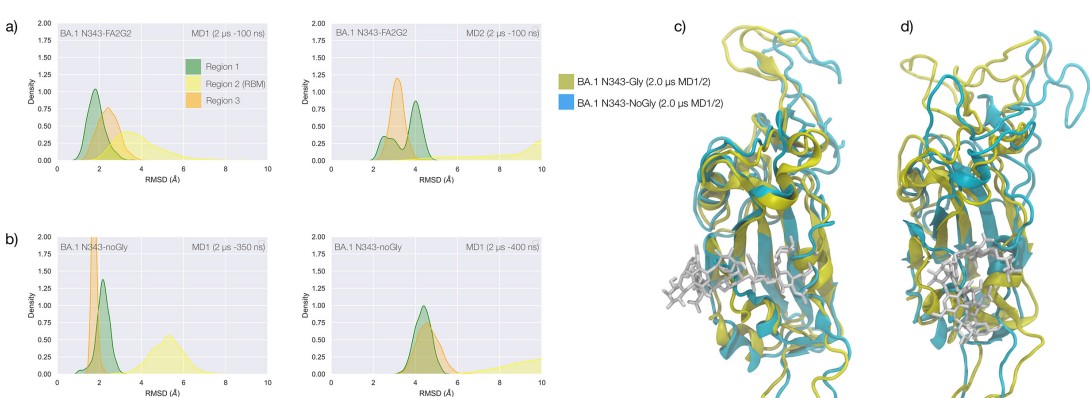

**Appendix 1—figure 6.** Conformational dynamics of the BA.1 receptor binding domain (RBD) in function of N343 glycosylation. (**a**) Kernel density estimates (KDE) plot of the backbone RMSD values calculated relative to frame 1 (t=0) of the MD1 (left) and MD2 (right) trajectories for Region 1 (green) aa 337–353, Region 2 (yellow) aa 439–506, and Region 3 (orange) aa 411–426 of the glycosylated omicron (BA.1) RBD. (**b**) KDE plot of the backbone RMSD values calculated relative to frame 1 (t=0) of the MD1 and MD2 trajectories (see details above) of the non-glycosylated omicron (BA.1) RBD. (**c**) Graphical representation of the structural alignment of the glycosylated (protein in yellow cartoons and N343-FA2G2 in white sticks, N331 omitted for clarity) and non-glycosylated (protein in cyan cartoons) of the omicron (BA.1) RBD from MD1. (**d**) Graphical representation of the structural alignment of the glycosylated (protein in yellow cartoons and N343-FA2G2 in white sticks, N331 omitted for clarity) and non-glycosylated (protein in cyan cartoons) of the omicron (BA.1) RBD from MD2. Structures correspond to the last frames of the trajectories, see details in the legend. Rendering done with VMD (https://www.ks.uiuc.edu/Research/vmd/) and KDE analysis with seaborn (https://seaborn.pydata.org/).

