## [Editor Report · eLife assessment]

This study presents an **important** finding on the structural role of glycosylation at position N343 of the SARS-CoV-2 spike protein's receptor-binding domain in maintaining its stability, with implications across different variants of concern. The evidence supporting the claims of the authors is **convincing**, since appropriate and validated methodology in line with current state-of-the-art has been approached. The work will be of interest to evolutionary virologists.

---

## [Referee Report · Reviewer #2 (Public Review)]

The authors sought to establish the role played by N343 glycosylation on the SARS-CoV-2 S receptor binding domain structure and binding affinity to the human host receptor ACE2 across several variants of concern. The work includes both computational analysis in the form of molecular dynamics simulations and experimental binding assays between the RBD and ganglioside receptors.

The work extensively samples the conformational space of the RBD beginning with atomic coordinates representing both the bound and unbound states and computes molecular dynamics trajectories until equilibrium is achieved with and without removing N343 glycosylation. Through comparison of these simulated structures, the authors are able to demonstrate that N343 glycosylation stabilizes the RBD. Prior work had demonstrated that glycosylation at this site plays an important role in shielding the RBD core and in this work the authors demonstrate that removal of this glycan can trigger a conformational change to reduce water access to the core without it. This response is variant dependent and variants containing interface substitutions which increase RBD stability, including Delta substitution L452R, do not experience the same conformational change when the glycan is removed. The authors also explore structures corresponding to Alpha and Beta in which no structure-reinforcing substitutions were identified and two Omicron variants in which other substitutions with an analogous effect to L452R are present.

The authors experimentally assessed these inferred structural changes by measuring the binding affinity of the RBD for the oligosaccharides of the monosialylated gangliosides GM1os and GM2os with and without the glycan at N343. While GM1os and GM2os binding is influenced by additional factors in the Beta and Omicron variants, the comparison between Delta and Wuhan-hu-1 is clear: removal of the glycan abrogated binding for Wuhan-hu-1 and minimally affected Delta as predicted by structural simulations.

In summary, these findings suggest, in the words of the authors, that SARS-CoV-2 has evolved to render the N-glycosylation site at N343 "structurally dispensable". This study emphasizes how glycosylation impacts both viral immune evasion and structural stability which may in turn impact receptor binding affinity and infectivity. Mutations which stabilize the antigen may relax the structural constraints on glycosylation opening up avenues for subsequent mutations which remove glycans and improve immune evasion. This interplay between immune evasion and receptor stability may support complex epistatic interactions which may in turn substantially expand the predicted mutational repertoire of the virus relative to expectations which do not take into account glycosylation.

---

## [Referee Report · Reviewer #3 (Public Review)]

Summary:

The receptor binding domain of SARS-Cov-2 spike protein contains two N-glycans which have been conserved the variants observed in these last 4 years. Through the use of extensive molecular dynamics, the authors demonstrate that even if glycosylation is conserved, the stabilization role of glycans at N343 differs among the strains. They also investigate the effect of this glycosylation on the binding of RBD towards sialylated gangliosides, also as a function of evolution

Strengths:

The molecular dynamics characterization is well performed and demonstrates differences on the effect of glycosylation as a factor of evolution. The binding of different strains to human gangliosides shows variations of strong interest. Analyzing structure function of glycans on SARS-Cov-2 surface as a function of evolution is important for the surveillance of novel variants, since it can influence their virulence.

Weaknesses:

The revised article does not hold significant weaknesses

---

## [Author Response]

The following is the authors’ response to the original reviews.

We are thankful to all reviewers and to you for your careful analysis of our work and for the feedback you all provided. The reviews were fundamentally positive with very minor modifications suggested, which we have addressed in this new version as follows.

We changed Figure 1 to include a high resolution image of the 3D structure of the low affinity complex between the RBD and the GM1 tetrasaccharide (GM1os), see panel d. We predicted this structure through extensive sampling through MD simulations as part of earlier work aimed at guiding the resolution of a crystal structure. Due to insurmountable difficulties in the crystallization of such complex the work was only published as an extended abstract(Garozzo, Nicotra, and Sonnino 2022). Following one of the reviewer’s suggestions we added all the details on the computational approach we used as Supplementary Material.We added the comment and corresponding references to the Discussion section in relation to earlier work flagged by one of the Reviewers (Rochman et al. 2022) “Further to this, our results show that taking into consideration the effects on _N-_glycosylation on protein structural stability and dynamics in the context of specific protein sequences may be key to understanding epistatic interactions among RBD residues, which would be otherwise very difficult, where not impossible, to decipher.”

References

Garozzo, Domenico, Francesco Nicotra, and Sandro Sonnino. 2022. “‘Glycans and Glycosylation in SARS-COV2 Infection’ Session at the XVII Advanced School in Carbohydrate Chemistry, Italian Chemical Society. July 4th -7th 2021, Pontignano (Si), Italy.” *Glycoconjugate Journal* 39 (3): 327–34.

Rochman, Nash D., Guilhem Faure, Yuri I. Wolf, Peter L. Freddolino, Feng Zhang, and Eugene V. Koonin. 2022. “Epistasis at the SARS-CoV-2 Receptor-Binding Domain Interface and the Propitiously Boring Implications for Vaccine Escape.” *MBio* 13 (2): e0013522.